# A Review of Multi-Sensor Fusion in Autonomous Driving

**DOI:** 10.3390/s25196033

**Published:** 2025-10-01

**Authors:** Hui Qian, Mingchen Wang, Maotao Zhu, Hai Wang

**Affiliations:** 1School of Automotive Rngieering, Nantong Institute of Technology, Nantong 226002, China; 20120011@ntit.edu.cn (H.Q.); zhumt@ujs.edu.cn (M.Z.); 2School of Automotive and Traffic Engineering, Jiangsu University, Zhenjiang 212013, China; wanghai1019@163.com

**Keywords:** camera–LiDAR fusion, multi-sensor fusion, object detection, deep learning, autonomous driving

## Abstract

Multi-modal sensor fusion has become a cornerstone of robust autonomous driving systems, enabling perception models to integrate complementary cues from cameras, LiDARs, radars, and other modalities. This survey provides a structured overview of recent advances in deep learning-based fusion methods, categorizing them by architectural paradigms (e.g., BEV-centric fusion and cross-modal attention), learning strategies, and task adaptations. We highlight two dominant architectural trends: unified BEV representation and token-level cross-modal alignment, analyzing their design trade-offs and integration challenges. Furthermore, we review a wide range of applications, from object detection and semantic segmentation to behavior prediction and planning. Despite considerable progress, real-world deployment is hindered by issues such as spatio-temporal misalignment, domain shifts, and limited interpretability. We discuss how recent developments, such as diffusion models for generative fusion, Mamba-style recurrent architectures, and large vision–language models, may unlock future directions for scalable and trustworthy perception systems. Extensive comparisons, benchmark analyses, and design insights are provided to guide future research in this rapidly evolving field.

## 1. Introduction

Autonomous driving has emerged as one of the most transformative and disruptive technologies in the modern transportation landscape. It aims to significantly reduce traffic accidents, improve fuel efficiency, and enhance mobility services. As intelligent vehicles must operate in highly dynamic and complex environments, the capability to perceive, understand, and interact with surroundings in real time is paramount. This necessitates not only the use of powerful perception algorithms but also the integration of information from multiple sensing modalities, giving rise to multimodal sensor fusion as a core component in autonomous driving systems.

Multimodal fusion refers to the process of combining data from different sensors to build a more comprehensive and reliable representation of the environment. In the context of autonomous driving, typical sensor suites include cameras, LiDARs, Radars, ultrasonic sensors, GPS, and inertial measurement units (IMUs). Cameras provide dense semantic information and fine-grained texture, while LiDARs generate accurate 3D point clouds for geometric structure, and Radars offer robust range and velocity measurements under adverse weather. However, each sensor is limited in isolation—cameras suffer in low light or fog, LiDARs are expensive and sensitive to weather conditions, and Radars produce sparse data. Therefore, fusing these modalities not only compensates for individual weaknesses but also enhances perception robustness.

In recent years, extensive efforts have been made to design efficient and scalable fusion frameworks that can integrate multimodal data at different abstraction levels. Fusion strategies are typically categorized into early fusion (sensor-level), mid-level fusion (feature-level), and late fusion (decision-level). Early fusion operates directly on raw data but suffers from misalignment and synchronization challenges. Mid-level fusion, often adopted in modern systems, combines intermediate features extracted from modality-specific backbones, enabling flexible attention mechanisms and alignment networks. Late fusion combines decisions from modality-specific branches, offering simplicity but often at the cost of reduced synergy.

Representative works have demonstrated the power of fusion in improving both perception and planning. For example, BEVFusion [1] constructs a unified Bird’s Eye View (BEV) representation by aligning multi-sensor features spatially, supporting multi-task learning such as 3D object detection and segmentation. TransFuser [2] introduces a mid-level fusion mechanism where camera and LiDAR features are fused through cross-attention modules. DeepFusion [3] leverages deep multi-scale feature fusion to address occlusion and sparsity in urban driving scenes.

Additionally, novel fusion paradigms have emerged, including Transformer-based architectures, BEV-centric frameworks, and temporal fusion for dynamic reasoning. M2BEV [4] introduces multi-scale BEV fusion, while LMDrive [5] incorporates temporal memory and vision–language alignment for robust planning. The rise of data-driven Transformer models like FusionAD [6] and MTR [7] further expands the potential of scalable, end-to-end multimodal fusion.

However, the development of robust fusion methods remains challenging. Sensor degradation due to hardware aging or environmental factors, modality misalignment during calibration or motion, and domain shifts across geographic or weather conditions can all affect performance. Furthermore, the scarcity of labeled multimodal datasets, especially for edge cases, hinders the training and generalization of fusion models. Real-time deployment constraints also demand efficient architectures that balance accuracy with latency and energy consumption.

In this review, we primarily focus on publications from top IEEE venues between 2020 and 2024, including CVPR, ICCV, ECCV, ICRA, IV, and TITS. Studies from the literature were retrieved from IEEE Xplore, arXiv, and Google Scholar using keywords such as “multi-modal sensor fusion”, “BEV representation”, “camera–LiDAR fusion”, and “multi-sensor autonomous driving”. We emphasize peer-reviewed works with innovative architectures and benchmark performance (e.g., nuScenes Leaderboard). References from robotics or agriculture are cited as inspirational cases and are not included in the core technical comparison. A summary of included works by domain and time span is provided in Table 1 to ensure reproducibility.

To provide a comprehensive understanding of this evolving field, this survey presents a structured and in-depth review of multimodal sensor fusion for autonomous driving. The primary goals of this work are as follows:To introduce the theoretical foundations behind sensor fusion, including mathematical models, uncertainty handling, and fusion principles.To analyze and categorize major fusion strategies based on the abstraction level, target tasks, and architectural design.To discuss the advantages and limitations of camera–LiDAR fusion, BEV transformation, cross-modal Transformer layers, and temporal fusion methods.To highlight real-world deployment challenges, including robustness under sensor failure, calibration errors, and domain generalization.To explore inspirations from other domains, particularly agriculture and robotics, for understanding the transferability of fusion architectures.To envision future directions with emerging paradigms such as foundation models, diffusion-based representation recovery, and large language model (LLM)-driven decision making.

## 2. Theoretical Foundations and Sensor Characteristics

### 2.1. Theoretical Foundations of Sensor Fusion

The theoretical foundation of multimodal sensor fusion in autonomous driving builds upon the confluence of estimation theory, information theory, and deep representation learning. At the heart of the fusion problem lies the objective to reconstruct or estimate the underlying true state of the environment, denoted as latent variables *x*, given a set of noisy and possibly redundant observations from multiple sensors $\left\{z_{1},\;z_{2},\;...,\;z_{M}\right\}$. The fusion process thus becomes a task of computing the posterior distribution $p\left(x\;|\;z_{1},\;...,\;z_{M}\right)$.

#### 2.1.1. Bayesian Filtering and Probabilistic Estimation

Bayesian estimation provides the classical foundation for fusing heterogeneous data. Under this framework, each sensor contributes a likelihood term $p\left(z_{i}\;|\;x\right)$, and the posterior is updated through recursive filtering:(1)$$p(x_{t}|z_{1:t})\propto p(z_{t}|x_{t})\int p(x_{t}|x_{t-1})p(x_{t-1}|z_{1:t-1})dx_{t-1}$$

Common implementations include the Kalman Filter (KF) for linear-Gaussian systems, the Extended KF (EKF) and Unscented KF (UKF) for nonlinear systems, and Particle Filters (PF) for nonparametric estimation. These models are extensively used for sensor fusion in localization, tracking, and SLAM systems by integrating GPS, IMU, LiDAR, and visual odometry data [8].

#### 2.1.2. Multi-View Learning and Representation Fusion

In deep learning-based fusion, each modality $z_{i}$ is processed by a modality-specific encoder $f_{i}$ to extract latent representations $h_{i}=\;f_{i}\left(z_{i}\right)$. Fusion mechanisms then combine $\left\{h_{1},\;...,\;h_{M}\right\}$ into a joint representation $h_{f}$, which serves as input to a downstream decoder. Fusion strategies include:Concatenation or summation (early fusion).Cross-modal attention (mid fusion).Weighted ensemble or voting (late fusion).

The theoretical insight here lies in leveraging conditional independence or conditional correlation among sensors to improve the representation power. For instance, TransFuser [2] uses cross-attention between camera and LiDAR features to implicitly learn inter-modal dependency.

#### 2.1.3. Uncertainty Modeling in Sensor Fusion

A critical challenge in real-world fusion is modeling sensor uncertainty. Probabilistic deep learning methods like Monte Carlo Dropout, deep ensembles, and variational inference capture epistemic and aleatoric uncertainties. Heteroscedastic regression losses are used when different sensors have varying noise levels. DeepFusion [3] and VAD [9] integrate uncertainty estimation modules to suppress noisy modalities in challenging conditions.

Information-theoretic formulations also arise in fusion: minimizing mutual information $I\left(h_{i};\;x\;|\;h_{j}\right)$ encourages the network to exploit complementary modalities, while bottleneck objectives (e.g., $I(h_{f};x)-\beta \sum I(h_{f};z_{i})$) control redundancy.

#### 2.1.4. Transformer-Based Fusion and Token Alignment

Recent works adopt Transformer-based architectures where modality embeddings are treated as tokens and fused via self-attention. The attention mechanism can be interpreted as a soft alignment that encodes relevance between modalities, closely related to belief propagation in graphical models. TransFusion and FusionAD apply token-level fusion in BEV space, leveraging spatial consistency and attention-based context aggregation.

In summary, the theoretical tools that govern sensor fusion span probabilistic graphical models, multi-view representation learning, uncertainty-aware inference, and Transformer-based attention alignment. These perspectives jointly empower the fusion of multimodal sensor data into coherent and robust perception frameworks for autonomous driving.

In the next subsection, we provide a detailed comparison of sensing modalities and analyze their complementary fusion properties.

### 2.2. Sensing Modalities and Fusion Properties

Effective multimodal fusion in autonomous driving critically depends on the characteristics, strengths, and limitations of individual sensing modalities. A thorough understanding of these properties allows researchers to design fusion systems that exploit complementary information and mitigate sensor-specific weaknesses. This section provides a detailed examination of commonly used sensors in autonomous vehicles—cameras, LiDARs, Radars, ultrasonic sensors, IMUs, and GNSS, which are shown in Table 2 and discusses how their attributes impact fusion strategies.

#### 2.2.1. Sensors

Cameras are the most prevalent and cost-effective sensing modality in autonomous driving. They provide high-resolution, dense 2D visual data, rich in semantic context such as object appearance, traffic signs, lane markings, and texture. Stereo and monocular configurations are widely deployed, and fisheye or surround-view systems enhance coverage. In fusion systems, cameras are typically used to provide fine-grained object classification and contextual understanding. However, their 3D spatial accuracy is limited without additional geometric priors.

LiDAR (Light Detection and Ranging) sensors provide sparse but highly accurate 3D point clouds representing the geometric structure of the environment. Rotating, solid-state, and flash LiDARs are all used in practice, varying in field-of-view, resolution, and cost. LiDARs are indispensable for tasks requiring precise 3D localization and obstacle detection. Fusion with camera data enables semantic enrichment of geometric data, as seen in [2,4].

Radar sensors emit radio waves and analyze reflections to measure range and relative velocity. They offer long-range detection and perform robustly under adverse weather conditions. Radars are typically fused with LiDAR or camera data to improve detection stability, especially for moving objects. Their temporal coherence is valuable for multi-frame fusion and tracking.

Ultrasonic sensors use sound waves to detect nearby objects, primarily for low-speed maneuvering tasks such as parking. Due to their narrow application scope, ultrasonic data is rarely used in high-level fusion tasks but may contribute to low-level safety functions.

Inertial Measurement Units (IMUs) provide acceleration and angular velocity data, essential for short-term motion estimation and ego-motion correction. IMUs are often fused with GPS and visual odometry in Extended Kalman Filter (EKF) or learning-based SLAM systems [8].

Global Navigation Satellite Systems (GNSS) (e.g., GPS, BeiDou) delivers global position estimates, critical for large-scale navigation. GNSS is typically fused with IMU and map data for coarse localization, while LiDAR or visual systems handle fine-grained positioning.

#### 2.2.2. Complementarity in Sensor Fusion

Each modality contributes unique information—semantic richness from cameras, spatial geometry from LiDARs, motion priors from Radar, inertial clues from IMUs, and global coordinates from GNSS. Effective fusion designs account for:Cross-modal redundancy (e.g., camera and LiDAR overlap).Temporal complementarity (e.g., Radar providing velocity between LiDAR frames).Robust fallback paths under sensor degradation (e.g., Radar-only detection under heavy fog).

For instance, BEVFusion [1] projects all modalities into a unified BEV space to achieve spatial alignment and joint reasoning.

In summary, the heterogeneity of sensing modalities enriches autonomous driving systems with multi-faceted environmental understanding. A deep appreciation of their fusion potential and inherent limitations guides the architectural choices in modern multimodal frameworks.

### 2.3. Learning Paradigms for Fusion

Modern sensor fusion techniques in autonomous driving increasingly rely on data-driven learning paradigms to extract, align, and integrate features from diverse sensing modalities. The learning framework not only determines how modalities are combined, but also affects the model’s ability to generalize, adapt, and scale to real-world complexities. This section introduces three major learning paradigms employed in multimodal fusion: deep learning-based fusion, probabilistic fusion models, and hybrid or self-supervised approaches. The compare between the representative fusion methodology is shown in Table 3.

#### 2.3.1. Deep Learning Based Senor Fusion

Deep learning has emerged as the dominant paradigm in sensor fusion due to its capacity to learn complex feature hierarchies and cross-modal correlations. Convolutional neural networks (CNNs), graph neural networks (GNNs), and transformer architectures are widely used to fuse visual, spatial, and temporal information.

For example, Transfuser [2], Interfuser [10], and BEVFusion [1], respectively, employ fusion within CNN backbones, interpretable transformer blocks, and BEV-aligned multi-task heads. For instance, Transfuser employs early-stage camera and LiDAR fusion within a shared CNN backbone, followed by a multi-branch transformer decoder that separately processes route planning and control signals. Similarly, M2BEV [4] projects multi-modal inputs into BEV space and uses attention-based modules to model inter-modal consistency and contextual interactions. These methods exploit the representation power of deep networks to directly learn fusion strategies from data.

Key advantages include:Scalability to high-dimensional sensory data.End-to-end optimization from raw input to control output.Ability to capture nonlinear and long-range cross-modal dependencies.

However, deep models are often criticized for their lack of interpretability and reliance on large labeled datasets. Overfitting to training domains and sensitivity to sensor failure also remain challenges.

#### 2.3.2. Probabilistic Fusion Models

Probabilistic models offer a principled framework to handle uncertainty, sensor noise, and partial observability. Techniques such as Kalman Filters, Bayesian networks, and Particle Filters have traditionally been used in localization and tracking tasks. These models estimate the posterior distribution of state variables given noisy observations from multiple sensors.

For instance, in self-driving applications, LiDAR-based pose estimations are fused with IMU and GNSS data through an Extended Kalman Filter (EKF) to yield robust localization [8].

Advantages of probabilistic methods include:Explicit modeling of noise and confidence in sensor readings.Smooth state estimation over time.Lightweight implementations suitable for real-time systems.

Nonetheless, their performance heavily depends on accurate sensor noise models and hand-crafted dynamics, limiting flexibility in highly dynamic or unstructured environments.

#### 2.3.3. Hybrid and Self-Supervised Approaches

To overcome the limitations of purely supervised learning or probabilistic methods, recent works have explored hybrid paradigms. These systems integrate neural network-based perception with probabilistic inference or leverage self-supervised objectives to enhance data efficiency.

FusionAD [6] performs multi-task sensor fusion by integrating perception, prediction, and planning within a BEV-fusion architecture. The model employs unified supervision across object detection, lane topology, and planning heatmaps, allowing better spatial-temporal reasoning. Similarly, HydraFusion [11] proposes a selective fusion strategy that dynamically adjusts the fusion level (early, intermediate, or late) based on scenario context. The model is supervised at multiple levels (e.g., object detection, affordance prediction, trajectory forecasting), enabling robust end-to-end learning from sparse annotations.

Other approaches exploit self-supervised signals, such as temporal consistency or cross-modal reconstruction. For instance, the MotionNet pipeline [12] jointly learns BEV features from LiDAR and camera streams by enforcing consistency between predicted future frames and actual sensor readings.

These hybrid strategies provide:Improved generalization under label scarcity.Robustness to partial sensor failure.

#### 2.3.4. Opportunities for Online Adaptation and Continual Learning

Learning paradigms shape the capabilities of sensor fusion frameworks in fundamental ways. Deep learning models enable high-capacity fusion pipelines but demand large annotated datasets. Probabilistic methods offer robustness and interpretability but are often inflexible. Hybrid and self-supervised techniques bridge the gap, offering a promising path forward by combining learning efficiency with resilience.

## 3. Fusion Architectures and Methodologies

### 3.1. Fusion Strategies

As Figure 1 shows, Fusion strategies in autonomous driving systems are commonly categorized into three stages: early fusion, mid-level fusion, and late fusion. Each stage reflects the timing and abstraction level at which multi-modal sensor data, typically from cameras, LiDAR, Radar, and sometimes IMU or GPS, is integrated within the processing pipeline.

#### 3.1.1. Early Fusion

Early fusion combines raw or minimally processed data from multiple sensors before any substantial feature extraction. The motivation is to preserve the rich information content across all modalities, allowing a neural network to learn joint features from the very beginning. For instance, raw LiDAR point clouds may be projected into image space and concatenated with RGB pixels, as seen in approaches such as [13], where candidate-level fusion leverages camera and LiDAR information jointly at the input.

An advantage of early fusion lies in its potential to learn deeply coupled cross-modal interactions. It allows the network to leverage complementary modalities early on—for example, aligning semantic texture from the camera with structural geometry from LiDAR. However, early fusion also introduces significant challenges: raw data from different sensors often differ in spatial resolution, coordinate frames, sampling rates, and noise characteristics. Aligning such data naively can inject ambiguity or redundancy. Moreover, early fusion tends to result in high-dimensional input spaces, requiring large models and extensive data to train effectively.

#### 3.1.2. Mid-Level Fusion

Mid-level fusion, by contrast, processes each modality independently to obtain semantically meaningful features before performing integration. This allows the network to exploit domain-specific feature extractors—e.g., CNNs tailored to images and PointNet++ or voxel-based encoders for point clouds—prior to fusion. BEVFusion [1] and Transfuser [2] exemplify this strategy by transforming both camera and LiDAR modalities into the BEV space, followed by joint fusion via attention or convolutional operations.

Mid-level fusion provides a flexible compromise: it reduces the dimensionality of inputs, mitigates sensor misalignment, and enables task-specific fusion heads for downstream components such as detection or planning. Furthermore, it accommodates missing or degraded sensor inputs by allowing fallback to single-modality streams. One drawback, however, is the potential loss of fine-grained cross-modal cues due to decoupled early processing. Cross-attention mechanisms are often used to compensate for this gap.

#### 3.1.3. Late Fusion

Late fusion strategies keep modalities separate through most of the network and only merge outputs at the decision level, such as combining detection results or trajectory predictions from each modality. While this approach lacks deep joint feature learning, it offers high modularity and robustness. For example, in [14], late fusion aggregates detections from multiple views to improve robustness in adverse conditions.

Late fusion is computationally efficient and interpretable, and allows independent optimization of each sensor’s processing pipeline. However, its main limitation is its inability to capture inter-modal dependencies at the feature level, making it suboptimal for tasks requiring rich spatial or semantic integration.

#### 3.1.4. Comparative Evaluation

In practice, many real-world autonomous driving systems adopt a hybrid fusion strategy. For instance, HydraFusion [11] dynamically adjusts the fusion point depending on context and task, exploiting early fusion in clear environments for detailed reasoning and shifting to mid or late fusion in adverse conditions for robustness. Similarly, MMF [15] employs mid-level fusion with cross-attention between BEV features and image tokens, achieving improved generalization under occlusions.

In summary, the choice of fusion timing: early, mid, or late, should be guided by sensor characteristics, computational constraints, and downstream task requirements. While early fusion enables deep inter-modal coupling, mid-level fusion provides a balance of flexibility and efficiency, and late fusion ensures safety-critical redundancy. Future research may explore adaptive fusion schemes that adjust dynamically based on context and task feedback. However, most current fusion systems adopt fixed-stage strategies that may not adapt well to varying environmental conditions. Early fusion is susceptible to noise propagation due to sensor misalignment, while mid-level fusion often lacks sufficient cross-modal granularity. Late fusion provides modularity but fails to capture joint semantics, especially under occlusion or partial observability.

### 3.2. Camera–LiDAR Fusion Architectures

Fusion between camera and LiDAR is a cornerstone of modern autonomous driving systems. These two modalities are fundamentally complementary: cameras provide rich semantic and texture information, while LiDAR offers accurate 3D geometric structure and depth. The challenge lies in effectively aligning and integrating these heterogeneous sources. In this section, we categorize Camera–LiDAR fusion architectures into two primary paradigms: query- or token-based alignment mechanisms and unified representation fusion frameworks, which is shown in Figure 2. Each paradigm presents unique design philosophies and trade-offs in terms of flexibility, accuracy, and deployment feasibility.

#### 3.2.1. Cross-Modal Token Alignment and Attention-Based Fusion

Recent advances in transformer-based architectures have enabled expressive cross-modal fusion using tokens and attention mechanisms. These approaches treat image and LiDAR features as tokenized sequences, allowing selective and spatially aware feature alignment through self- and cross-attention modules.

CMT [16] exemplifies this paradigm by encoding LiDAR and camera features into modality-specific token sequences. A cross-modality transformer then enables selective attention across these tokens, focusing on semantically important regions. This mechanism improves robustness against occlusions and facilitates flexible fusion across viewpoints.

Similarly, FusionAD [6] adopts a BEV-centric transformer where modality-aware tokens are embedded directly in BEV space, enabling unified reasoning across tasks like detection and planning. Hierarchical alignment modules refine token-level matching across depth and semantics, resulting in robust performance under degraded sensor conditions.

LiftFusion [17] proposes lifting multi-scale image and LiDAR features to a shared latent space, then applying deformable attention to account for spatial misalignments. This structure preserves both high-resolution details and global dependencies.

Other designs like UniM2Fusion [18] introduce token distillation strategies to compress multi-modal sequences into unified representations, reducing inference latency while maintaining fidelity. In this paradigm, the flexibility of attention allows fine-grained control over what and where to fuse, and the modularity of token pipelines enables task-specific adaptation.

Key advantages include:Modality-agnostic structure with adaptive interaction.Rich spatial and semantic alignment via attention weights.Compatibility with long-horizon reasoning and memory fusion.

However, challenges remain in managing token length, computational load, and alignment without extensive calibration. Future improvements may involve dynamic token pruning, online alignment learning, and reinforcement-guided attention for safety-critical scenarios.

#### 3.2.2. Unified Representation Fusion in BEV Space

An alternative to token-centric fusion is the construction of unified representations, typically in BEV space, where both camera and LiDAR features are projected and aggregated. This design emphasizes spatial alignment and global scene consistency.

BEVFusion [1] represents a seminal work in this direction. It unifies LiDAR voxels and lifted image features into BEV through geometric transformation, enabling spatially consistent joint processing using shared convolutional backbones.

M2BEV [4] improves upon this by adopting dual-path fusion with shared BEV tokens and attention for modality interaction. This structure maintains structured geometry while enabling modality-specific refinement, leading to better generalization across scenes and modalities.

AutoAlignV2 [19] addresses calibration misalignment by using deformable convolutions to dynamically learn spatial offsets during fusion. This approach relaxes the strict reliance on extrinsic parameters and enhances robustness under sensor noise.

Furthermore, methods like HVNet [20] and DeepInteraction [21] adopt dense multi-scale fusion in BEV space, integrating fine-grained and coarse features for hierarchical scene understanding. They often incorporate auxiliary tasks like depth completion or semantic segmentation to reinforce shared feature learning.

Unified representation approaches offer strong spatial priors, compact memory footprints, and natural compatibility with BEV-based planning. Their benefits include:Consistent geometry for downstream reasoning.Compatibility with CNN-based detectors and planners.Robustness to partial observations through spatial pooling.

However, these methods can be sensitive to projection quality and may lose fine texture details from images. Solutions involve multi-scale projection, semantic-aware lifting, and incorporation of uncertainty in projection.

In summary, Camera–LiDAR fusion architectures are evolving towards expressive, modular, and spatially grounded designs. The dual pathway of token-based interaction and BEV-unified representation forms the basis of current and future advances, combining flexibility, interpretability, and geometric consistency for robust multi-modal autonomous perception.

### 3.3. Multi-Modal BEV Construction

Multi-modal BEV construction represents a paradigm in autonomous driving where diverse sensor inputs are projected into a shared top-down spatial representation. Unlike query-aligned methods that rely on dynamic token attention, this category of approaches emphasizes geometric alignment through explicit projection operations, enabling structured fusion across LiDAR, camera, and Radar sensors. This section focuses exclusively on methods that construct a unified BEV representation prior to fusion, highlighting projection strategies, architectural integration, and spatial alignment mechanisms.

#### 3.3.1. Unified BEV Projection of Modalities

The foundation of this fusion class lies in projecting heterogeneous sensor data into a shared BEV coordinate system. Camera inputs, which inherently capture 2D perspective information, require lifting into 3D space. PETR [22] addresses this by estimating pseudo-depths and projecting image features into 3D voxel grids, which are subsequently discretized into BEV. Similarly, BEVFusion [1] independently lifts both camera and LiDAR features into BEV before fusing them via convolutional backbones.

For LiDAR, the projection is more direct. Voxelization is widely used, as seen in CenterPoint [23], where 3D points are mapped to a voxel grid and aggregated for BEV encoding. In VoxFormer [24], the voxel encoder directly converts LiDAR point clouds into spatial-aware features in BEV format, which are subsequently fused with camera-derived features also lifted to BEV.

#### 3.3.2. Spatial Alignment and Fusion in BEV

Once modalities are unified in BEV space, effective fusion depends on spatial correspondence. Sparse4D [25] introduces spatiotemporal voxel tensors where lifted camera and LiDAR features are accumulated over time, then sliced along the BEV plane. This spatial unification enables cross-modal reasoning with explicit geometric consistency.

FusionPainting [26] adopts a teacher-student strategy where the LiDAR features guide the lifting and alignment of camera features in BEV. The semantic supervision helps address occlusion and sparsity problems. CBGS [27] complements this by adjusting the sampling and fusion weights in BEV according to object class distributions, enhancing long-tail recognition.

#### 3.3.3. Modular and Scalable Fusion Pipelines

BEV-based methods are often modular by design. Each modality undergoes separate geometric lifting before fusion, allowing flexible sensor configurations. For example, UVTR [28] processes LiDAR and camera through independent encoders and projection heads, then combines them via BEV-aligned fusion blocks. Although UVTR includes transformers, its core structure aligns with BEV construction rather than query-based fusion.

This modularity supports scaling to additional inputs like HD maps or Radar, as seen in extensions of BEVFusion. The unified spatial grid simplifies integration, enabling late-stage fusion in shared convolutional heads or attention mechanisms over the BEV plane.

#### 3.3.4. Limitations and Considerations

Despite their structured nature, BEV construction methods face challenges. Depth estimation for camera lifting remains error-prone under adverse conditions. Furthermore, memory efficiency is a concern when lifting high-resolution images or accumulating spatiotemporal features. The rigidity of the BEV grid may also suppress modality-specific nuances.

Nevertheless, unified BEV representation remains a compelling direction for sensor fusion, particularly for tasks requiring spatial precision like planning and map-based localization. As future work, improving the geometric reliability of camera lifting and developing lightweight BEV encoders will be key to deploying these methods in real-time autonomous systems.

### 3.4. Transformer-Based Query-Aligned Fusion

Transformer-based sensor fusion has emerged as a dominant paradigm in recent multi-modal autonomous driving systems, enabling flexible and fine-grained cross-modal interaction through the use of learned attention mechanisms. In contrast to the spatially structured BEV fusion discussed in Section 3.3, these approaches rely on token-level alignment, using queries to retrieve or aggregate modality-specific information. This section focuses on fusion methods that employ transformer architectures to align and integrate sensor inputs, particularly in the context of camera and LiDAR data.

#### 3.4.1. Query-to-Modality Attention and Cross-Modal Sampling

One of the core designs in query-aligned fusion methods is the use of learnable queries to interact with modality-specific features. For instance, TransFuser [2] introduces a fusion transformer that uses the output of camera and LiDAR encoders as keys and values, while the ego-agent state or navigation signal serves as the query input. These learned queries attend to relevant spatial features from each modality, supporting dynamic feature selection and improved contextualization.

CMT [16] extends this approach by utilizing task-driven queries to perform conditional fusion, where a separate query token is allocated per downstream task (e.g., detection, planning). Each query token aggregates relevant information from the multi-modal input space through attention, enhancing task specificity.

BEVFormer++ [29] similarly leverages deformable attention to sample from multi-view image features using BEV queries. Unlike voxelization approaches, the spatial correspondence is learned via attention, offering greater flexibility in handling occlusions or perspective distortions.

#### 3.4.2. Temporal and Sequential Query Extension

Transformer-based fusion also allows for the modeling of temporal sequences through attention over token memories. In UVTR [28], although originally designed for BEV lifting, a query-based memory mechanism is incorporated where features from previous frames are attended via temporal BEV tokens. This facilitates information retention across time steps and improves tracking and continuity in dynamic environments.

Sparse4D [25] introduces a sparse 4D attention structure that enables dynamic queries to select spatiotemporal features from voxelized representations. While its input lifting resembles BEV fusion, the core interaction is driven by transformer queries, enabling adaptive selection over both space and time.

#### 3.4.3. Unified Query Spaces and Semantic Conditioning

Recent works have explored embedding all modality features into a unified latent token space, conditioned on semantics or tasks. MVFusion [30] employs multi-view camera BEV tokens that are aligned with LiDAR-derived tokens through cross-modal attention. The shared token space serves as the fusion bottleneck, enabling mutual refinement.

In DeepInteraction [21], cross-attention modules are conditioned on geometric priors, such as depth or ego-motion, to align tokens sampled from each modality. The transformer dynamically computes correspondence based on semantic and geometric cues, rather than fixed spatial projections.

#### 3.4.4. Comparison to BEV Fusion and Challenges

Query-aligned transformer fusion offers several advantages over rigid BEV construction:Dynamic Sampling: Tokens can attend to informative regions regardless of fixed grid structures.Cross-Task Flexibility: Query tokens can be tailored per task or per modality.End-to-End Learning: Attention weights can implicitly learn fusion relevance, bypassing hand-crafted projection steps.

However, challenges include:Computation Cost: Full self-attention across large feature maps is expensive.Stability: Query-token learning is sensitive to initialization and data imbalance.Interpretability: Attention maps may be hard to interpret in safety-critical contexts.

Nonetheless, transformer-based query fusion has demonstrated strong performance in complex scenes and long-range perception tasks. With further developments in efficient attention and semantic guidance, it is expected to remain a core strategy in future multi-modal fusion architectures.

## 4. Task-Specific Applications of Multi-Modal Sensor Fusion

To systematically analyze how multi-sensor fusion techniques are applied across different perception tasks in autonomous driving, this section provides a task-wise review organized around core functional components such as depth completion, object detection, segmentation, tracking, and online calibration. Prior to the technical analysis, we present two summary tables to contextualize the following discussions.

Table 4 offers a comparative overview of representative multi-modal autonomous driving datasets, including KITTI, nuScenes, Waymo, PandaSet, and RADIATE. The table highlights key attributes such as sensor configurations, task coverage, environmental diversity, and annotation richness, which influence the design and evaluation of fusion models.

Table 5 then outlines the primary challenges associated with multi-sensor fusion for each task, mapping these difficulties to representative solutions. By summarizing common fusion bottlenecks—such as projection inconsistency, semantic misalignment, temporal desynchronization, or modality degradation—this table helps to establish a structured perspective for the detailed task-wise discussions that follow.

### 4.1. Depth Completion

Depth completion, the process of transforming sparse depth measurements into dense and accurate depth maps, plays a fundamental role in autonomous driving perception. In real-world autonomous vehicle (AV) systems, LiDAR sensors typically produce sparse point clouds due to limited beam density, occlusions, and range constraints. Consequently, projecting these sparse points into the image domain yields depth maps with significant missing regions, which, if not properly reconstructed, can severely affect downstream tasks such as 3D object detection, semantic segmentation, and free space estimation [31]. The challenge lies in not only filling in these gaps but also preserving fine geometric details and ensuring global consistency with scene semantics.

Historically, depth completion methods evolved from traditional geometry-based techniques to data-driven deep learning approaches. Early methods relied on spatial interpolation techniques such as nearest-neighbor filling, bilateral filtering, and thin plate splines, which exploited local smoothness assumptions to propagate depth values. While computationally inexpensive, these methods failed to capture discontinuities at object boundaries and struggled in highly non-uniform sparse data distributions. Later, SLAM-based reconstruction introduced multi-view geometric constraints, leveraging camera motion to infer depth in missing regions. However, these methods were sensitive to odometry errors and could not operate robustly in dynamic environments where moving objects violated static-scene assumptions.

The advent of deep learning enabled image-guided depth completion, where convolutional neural networks (CNNs) or transformers exploit correlations between RGB texture and sparse depth to predict dense depth maps [32]. These methods can be broadly categorized into supervised, semi-supervised, and self-supervised paradigms.

Supervised methods leverage dense ground truth from high-resolution LiDAR or structured light scanners to train regression models. Loss functions typically combine an L1/L2 term for absolute depth accuracy with gradient-based smoothness regularization to preserve edges:(2)$$L_{\mathrm{depth}}=\lambda_{1}\|\hat{D}-D^{\mathrm{gt}}{\|}_{1}+\lambda_{2}\|\nabla \hat{D}-\nabla D^{\mathrm{gt}}{\|}_{1},$$
where $\hat{D}$ is the predicted depth map, $D^{\mathrm{gt}}$ is the ground truth, and ∇ denotes spatial gradient.

Self-supervised methods remove the dependency on dense ground truth by exploiting photometric consistency across stereo or monocular frames. The idea is to warp one image into another using the predicted depth and known camera intrinsics, and minimize the reconstruction error. This approach inherently encourages geometric correctness but can be degraded by illumination changes and occlusions.

A critical aspect of depth completion in multi-modal fusion is how the LiDAR and camera features are combined. A common strategy is early fusion, where sparse depth is projected into the image plane and concatenated with RGB channels, allowing a single encoder to process both modalities [33]. Alternatively, late fusion maintains separate processing streams for each modality and merges their high-level features via attention or cross-modality convolution. The choice between these approaches impacts computational efficiency, robustness to sensor failures, and the ability to exploit complementary information.

In recent years, methods such as sparse-to-dense convolution and depth-aware attention have emerged to address the issue of irregular LiDAR sampling. Sparse convolution operators process only valid depth pixels, significantly reducing wasted computation on missing entries. Depth-aware attention modules, inspired by transformer architectures, explicitly model the relationship between depth values and image semantics, allowing the network to focus on structurally relevant regions [9]. This design not only improves the completion accuracy but also benefits downstream fusion tasks, since the resulting depth maps preserve physically meaningful geometry.

Interestingly, similar principles of depth completion are found in other domains, such as precision agriculture, where multi-modal sensing is employed for crop monitoring and yield estimation. For example, agricultural robots often combine sparse LiDAR canopy scans with high-resolution multispectral imagery to estimate biomass volume and plant height [34]. The challenges—non-uniform point distributions, occlusions from leaves, and environmental noise—mirror those in urban AV scenarios. Cross-domain knowledge transfer is possible: in agriculture, adaptive interpolation guided by multispectral reflectance indices has been shown to improve canopy depth estimation, suggesting that spectral-semantic priors could similarly enhance AV depth completion in scenes with ambiguous object boundaries.

From a dataset perspective, most AV depth completion research benchmarks on KITTI Depth Completion [35], NYU Depth V2 [36], and increasingly on nuScenes [37] for more challenging settings. KITTI, for example, provides synchronized RGB images and sparse depth maps with around 30 k training samples. State-of-the-art methods such as DeepLiDAR [38], FusionNet [39], and TransDepth [40] report RMSE improvements of over 15% compared to early CNN baselines. These gains largely stem from improved cross-modality feature alignment and robust handling of LiDAR sparsity. Some methods explicitly model the projection geometry from 3D to 2D using:(3)$$u=\frac{f_{x}X}{Z}+c_{x},\;v=\frac{f_{y}Y}{Z}+c_{y},$$
where *X*, *Y*, *Z* are 3D LiDAR coordinates, *u*, *v* are pixel coordinates, and $f_{x},\;f_{y},\;c_{x},\;c_{y}$ are camera intrinsics. Accurate calibration in this projection is vital for avoiding misalignment artifacts that can cascade into perception errors.

A notable advancement is the integration of uncertainty estimation into depth completion models. Predicting a per-pixel confidence score allows the fusion module to weigh high-confidence depth values more heavily in downstream decision-making. This is particularly valuable in adverse weather, where LiDAR returns may be noisy or partially missing. In agricultural applications, uncertainty maps have been used to mask unreliable canopy depth estimates caused by sunlight interference [41,42,43,44], and similar masking can be applied to AV depth maps to prevent propagating erroneous geometry into planning modules.

In summary, depth completion has transitioned from purely geometric interpolation to sophisticated, learning-based multi-modal fusion pipelines that exploit semantic guidance, sparse geometry processing, and attention-based alignment. The incorporation of cross-domain strategies from agriculture and other robotic applications holds promise for further robustness improvements. As AV perception systems become more tightly integrated with planning and control, the accuracy and reliability of depth completion will remain a key determinant of safe and efficient autonomous navigation.

### 4.2. Dynamic Object Detection

Dynamic object detection aims to localize and classify moving agents such as vehicles, pedestrians, and cyclists. Fusion strategies vary from sequential pipelines—where one modality generates proposals refined by another—to fully end-to-end architectures. In 2D proposal–based sequential models, an image detector provides 2D bounding boxes, which are projected into 3D frustums for point cloud–based refinement, as in F-PointNet and its derivatives [45]. This approach benefits from mature 2D detection models but inherits their limitations, including sensitivity to occlusion and dependence on precise extrinsic calibration.

Feature-level fusion methods bypass explicit proposal projection by jointly processing image and point cloud features in shared representations [46]. MV3D and AVOD aggregate BEV LiDAR features with front-view image features at the region-of-interest level [45], while ContFuse introduces point-wise continuous convolutions to align features across modalities [47]. Recent transformer-based architectures, such as TransFusion, employ cross-attention mechanisms to adaptively weight features from each modality, achieving robustness in low-light or partially degraded LiDAR scenarios [32].

In agricultural robotics, dynamic detection of moving machinery or animals often faces similar multimodal integration challenges [41,48,49,50]. For instance, fusing visible and thermal camera streams enables the detection of livestock in low-illumination barns, improving safety in automated feeding systems [51]. Such cross-domain examples demonstrate the relevance of modality-adaptive fusion for detecting moving entities under constrained visibility [41,50,52,53,54,55,56,57,58,59].

### 4.3. Static Object Detection

Static object detection encompasses the identification of stationary roadway elements, including lane markings, traffic signs, and static obstacles. For lane and road surface detection, BEV fusion methods project both LiDAR and camera data into a top-down grid to preserve geometric fidelity [1]. Multi-stage fusion strategies have been shown to improve maximum F1 scores on urban marked road benchmarks, particularly when integrating multi-scale features from both modalities [33].

Traffic sign recognition pipelines often detect candidate regions in LiDAR based on retro-reflectivity, project them onto the image plane, and classify them via deep CNNs [60]. Such result-level fusion is effective in leveraging the precise localization of LiDAR and the rich appearance cues of imagery. In agriculture, comparable methods detect fruit clusters or disease-affected regions by segmenting 3D point clouds of crops and then classifying projected image patches, benefiting from high-precision 3D localization and rich 2D spectral data [55].

### 4.4. Semantic and Instance Segmentation

Semantic segmentation assigns a class label to each pixel or point, while instance segmentation further differentiates individual object instances within the same class. Multi-modal fusion significantly enhances segmentation accuracy, especially for small or partially occluded objects. In 3D semantic segmentation, voxel-based fusion networks such as 3DMV project multi-view image features into 3D space, fusing them with voxelized point cloud features for per-voxel prediction [61]. Point-based fusion methods, such as MVPNet, directly associate image semantics with 3D points, improving fine-grained boundary delineation [62].

In agricultural sensing, semantic segmentation of crops, soil, and weeds benefits from combining multi-spectral image features with LiDAR-derived canopy structure [63]. This enables precise differentiation between plant species and background, even under overlapping foliage [46]. The shared challenge in both domains is aligning dense appearance data with sparse geometric measurements, where mid-level feature fusion proves most effective [64,65,66].

### 4.5. Multi-Object Tracking

Multi-object tracking (MOT) in autonomous driving relies on associating detections over time to maintain object identities. Tracking-by-detection frameworks typically use camera–LiDAR fusion for robust 3D localization and re-identification, with association performed via data association algorithms or learned affinity metrics [67]. Detection-free tracking approaches integrate fused features directly into filtering frameworks, such as labeled multi-Bernoulli filters, enabling continuous state estimation even without explicit detections [68].

In agriculture, MOT-like tracking is applied to monitor moving entities such as automated tractors or roaming livestock [69,70,71,72,73], where fused RGB, which is thermal data, improves tracking continuity in occluded or variable-light environments [74]. The capacity to sustain track integrity under partial sensor degradation is a common requirement across domains.

### 4.6. Online Cross-Sensor Calibration

Maintaining accurate calibration between sensors is essential for effective fusion, as even minor extrinsic drift can degrade perception performance. Deep learning–based calibration networks, such as RegNet, estimate extrinsics from raw RGB and depth features, while self-supervised methods like CalibNet minimize photometric and geometric inconsistencies between miscalibrated and target depth maps [75]. The main challenge lies in achieving real-time performance without sacrificing calibration accuracy.

In agriculture, online calibration has been applied to multi-camera rigs and LiDAR–camera systems mounted on field vehicles, compensating for mechanical shifts induced by rough terrain [76]. Such experiences emphasize the importance of continuous calibration for long-duration operations in variable environmental conditions, a requirement equally critical for autonomous road vehicles [70,76,77,78].

## 5. Current Challenges in Multi-Modal Fusion Perception for Autonomous Driving

Despite the remarkable progress in multi-modal fusion techniques for autonomous driving perception, significant challenges persist in bringing these methods to robust, scalable deployment under real-world constraints. The fusion of heterogeneous sensors—commonly including cameras, LiDARs, Radars, and occasionally ultrasonic or thermal imagers—encounters complex issues across multiple dimensions: sensor degradation, temporal synchronization, geometric misalignment, domain generalization, computational cost, interpretability, and training data scarcity. This chapter provides a comprehensive examination of the most critical challenges facing modern multi-sensor fusion perception systems and explores research efforts that aim to mitigate these limitations.

To ensure the interpretability of Figure 3, we define the scoring rationale for each axis in the radar chart as follows. Modality Robustness reflects the diversity and resilience of sensor fusion under occlusions or adverse weather, measured through ablation studies on nuScenes or RADIATE. Generalization captures the capacity to transfer across domains or tasks. Alignment Accuracy assesses the quality of geometric and semantic fusion across modalities. Temporal Consistency measures stability in sequential outputs and the presence of recurrent or state-space structures. Interpretability evaluates whether the model provides explainable outputs such as attention maps or language feedback. Scalability refers to the architectural modularity that facilitates extension to new sensors or tasks. These scores are derived from published model behaviors, qualitative ablations, and structural design principles.

### 5.1. Sensor Degradation and Failure Robustness

One of the most fundamental challenges in multi-modal fusion arises from geometric misalignment among sensors. This misalignment stems from both hardware calibration errors and intrinsic differences in field-of-view, resolution, and data sparsity across sensors. For instance, camera pixels and LiDAR points represent different sampling densities and coordinate frames, and aligning them accurately requires precise intrinsic and extrinsic calibration matrices. Even slight calibration drift due to mechanical vibrations or thermal shifts can result in significant fusion errors in dense prediction tasks such as segmentation and detection.

While projection-based methods (e.g., projecting LiDAR points into the image plane) are commonly used for early fusion, they are inherently limited by resolution mismatches. Conversely, late fusion strategies that maintain separate processing pipelines for each modality reduce reliance on exact alignment but face challenges in achieving coherent spatial correlation across the modalities. Hybrid strategies using BEV representations attempt to reconcile these differences, yet BEV warping may lose fine-grained semantic detail. Moreover, dynamic objects present additional difficulty, as their positions may change between the asynchronous acquisition timestamps of different sensors, causing further inconsistencies in projected representations.

To alleviate misalignment, methods such as geometric warping, learned spatial calibration, and cross-modal attention mechanisms have been proposed. For example, CrossFuse [47] and CM-KD [79] employ attention-guided spatial realignment, improving consistency between modalities. Transformer-based models like VPFNet [80] adopt global alignment via learnable token correspondence, though their scalability under high-resolution settings remains a concern. Future research may benefit from investigating self-calibrating frameworks that combine online extrinsic adjustment with semantic feedback, particularly under long-term deployments where sensor mounting parameters may drift.

Another emerging direction is leveraging uncertainty estimation during spatial alignment. Models such as ADF [81] incorporate uncertainty-aware feature fusion to prioritize alignment across high-confidence regions. Integrating such probabilistic representations with dynamic registration networks presents a promising pathway toward robust geometric consistency.

Lastly, temporal misalignment compounds geometric misalignment, particularly under ego-motion. Asynchronous sensor timestamps can cause positional drift when fusing dynamic object trajectories. Time-compensated BEV projection and flow-guided warping, such as proposed in DETR [82], mitigate these effects by interpolating motion states across sensor frames. Ensuring accurate alignment in these settings is vital for safe and coherent scene understanding.

### 5.2. Temporal and Spatial Misalignment

Multi-sensor fusion necessitates precise alignment in both space and time. In practice, due to asynchronous sampling rates, timestamp drift, and physical displacements of sensors, achieving tight synchronization is nontrivial. Misalignment between LiDAR scans and camera frames can introduce ghosting, duplicate edges, or shifted object representations, which negatively impact fused feature quality.

Several approaches attempt to mitigate this. For instance, pixel-level fusion methods apply temporal interpolation to match LiDAR timestamps with camera frames, while BEV-based methods employ ego-motion compensation to align sensor readings to a unified coordinate frame. Moreover, spatio-temporal attention mechanisms within fusion transformers can implicitly model alignment uncertainties by learning correspondence patterns across sensor modalities.

However, misalignment is often scene-dependent and can vary with vehicle velocity, vibration, or terrain irregularity. Online calibration networks like CalibNet [75] attempt to dynamically adjust extrinsics using photometric and geometric feedback, but real-time, drift-free operation remains elusive.

Future directions involve the development of fusion networks that explicitly model time and geometry as variables, such as using 4D occupancy grids or flow-based spatial alignment. There is also interest in integrating IMU data to improve motion compensation, especially during high-speed maneuvers or when visual cues are ambiguous. Cross-modal supervision strategies, where one sensor acts as a temporal reference for another, may help to better enforce consistency across frames.

In addition, synthetic datasets with controlled temporal offsets can be used to systematically train fusion models that are robust to synchronization issues. Domain knowledge from fields such as robotics and AR/VR, where temporal alignment is critical, could inform new alignment strategies. Finally, hardware-software co-design that emphasizes time-synchronized data capture and fast calibration routines will be crucial for ensuring real-time applicability of fusion perception systems.

### 5.3. Domain Generalization and Scene Diversity

Models trained on one geographic or environmental domain often perform poorly when deployed elsewhere due to shifts in sensor characteristics, scene layouts, weather patterns, or traffic participants. Fusion architectures tightly coupled to specific sensor configurations (e.g., number of LiDAR beams, camera FOV, or intrinsic resolution) are especially brittle under domain shift. These limitations severely constrain the scalability and generalizability of perception systems in real-world deployments.

To enhance domain robustness, numerous approaches have been explored. One common line of work involves domain adaptation via adversarial training, where the model learns to align feature distributions between source and target domains. In the context of multi-modal fusion, this may entail aligning LiDAR and camera features across domains using shared encoders or projection networks, as demonstrated in CM-KD [79]. Some methods incorporate cycle-consistent adversarial loss to retain modality-specific semantics while bridging domain gaps.

Style transfer-based data augmentation is another popular strategy. For example, methods that convert clear weather images into foggy or rainy conditions can enhance robustness across visibility conditions. Synthetic-to-real adaptation using photorealistic simulators such as CARLA [83] or Waymo OpenSim allows fusion models to pretrain on diverse conditions before fine-tuning on real data.

Contrastive learning has also emerged as a powerful paradigm for domain generalization. By encouraging modality-invariant representations through positive and negative pairs sampled across domains, fusion networks can learn to separate task-relevant semantics from domain-specific variations. Some works further disentangle geometry and appearance priors in BEV space, enabling generalization across unseen scenes or lighting.

In the agricultural robotics domain, cross-field adaptation has been tackled via meta-learning and few-shot adaptation, allowing models to transfer across different crop types, terrain structures, and environmental conditions. The study in [74] illustrates how segmentation models can be adapted with limited samples and minimal retraining. Drawing inspiration from this, AV fusion systems can benefit from parameter-efficient transfer mechanisms such as lightweight adapters, LoRA modules, or frozen-backbone finetuning for rapid deployment in new cities or under new sensor setups.

Moreover, the construction of large-scale, geographically diverse datasets is crucial. Initiatives like nuScenes, Lyft Level 5, and PandaSet have expanded data coverage across weather, cities, and day/night cycles. Fusion networks trained on such heterogeneous data are inherently more robust. However, ensuring balance and diversity in training remains a challenge, necessitating intelligent sampling and dataset curation techniques.

Finally, model-agnostic ensembling and Bayesian inference techniques may help quantify and manage domain uncertainty. By incorporating uncertainty estimates in fusion modules, systems can better identify distribution shifts and fall back to conservative policies when confidence is low—ensuring safety even under domain drift.

### 5.4. Computational Burden and Real-Time Constraints

Beyond robustness, practical deployment of multi-modal fusion systems faces stringent real-time and computational constraints. Autonomous driving platforms must process high-bandwidth sensor streams from LiDAR, cameras, and radars at frame rates of 10–30 Hz, while simultaneously performing detection, tracking, and planning within a few hundred milliseconds. This requirement places severe stress on both model design and hardware resources.

A key challenge lies in the computational overhead of deep fusion architectures. Transformer-based fusion networks, though powerful, often incur quadratic complexity with respect to input tokens, making them difficult to deploy on resource-limited onboard units. Similarly, voxel- or point-based 3D backbones require substantial GPU memory and FLOPs, which can exceed the capabilities of embedded automotive hardware.

Several strategies have been proposed to mitigate these constraints. Model compression techniques such as pruning, quantization, and knowledge distillation have been used to reduce parameter counts and latency without sacrificing accuracy. Lightweight fusion backbones leveraging MobileNet or PointPillars structures offer improved efficiency, albeit sometimes at the cost of robustness. Recently, state-space models such as Mamba have shown promise by replacing expensive attention operations with linear recurrence mechanisms, enabling long-sequence modeling at reduced complexity.

In addition, sensor-adaptive processing can further optimize resource usage. For instance, dynamically adjusting LiDAR beam density or camera resolution under favorable conditions can reduce redundant computations. Event-driven or asynchronous fusion pipelines have also been explored, where high-frequency sensors update critical states while low-frequency modalities provide complementary context.

Another promising direction is hardware-software co-design. Dedicated accelerators (e.g., TensorRT, FPGA-based fusion modules) enable real-time inference with significantly lower latency and power consumption. Integration with onboard domain controllers ensures that safety-critical tasks such as perception and planning are prioritized under strict time budgets.

Finally, balancing real-time constraints with safety demands necessitates graceful degradation mechanisms. When computational resources are saturated, fallback strategies such as using single-modality perception or coarser-grained BEV features allow the system to maintain functional safety, even if performance is reduced. Coupled with uncertainty-aware decision modules, this ensures that autonomous vehicles can continue operating safely without violating latency requirements.

### 5.5. Interpretability and Trustworthiness

Multi-modal fusion models are often treated as black boxes, raising concerns about interpretability and safety validation. When incorrect predictions arise—e.g., missed pedestrians or hallucinated obstacles—it is difficult to pinpoint whether the error stemmed from sensor noise, fusion failure, or representation collapse.

Explainable AI (XAI) techniques have started to be applied in this domain, such as saliency maps for multi-modal inputs or feature attribution for fusion layers. Techniques like Grad-CAM can highlight regions in the input sensors that most influenced a model’s decision, while Shapley values can estimate sensor-wise contribution to predictions.

Explainability has emerged as a critical design factor in multi-modal fusion frameworks for autonomous driving. As these models increasingly serve as black-box components within end-to-end driving systems, ensuring interpretability, transparency, and diagnosability is vital for safety validation and failure analysis.

Current explainability techniques can be broadly categorized into two groups:

Post hoc interpretability, which analyzes trained models using saliency methods such as Grad-CAM, guided backpropagation, or attention rollout. These techniques provide visual justifications of which modalities or spatial regions influenced a decision, but often suffer from limited fidelity or inconsistent semantics.

Intrinsic interpretability, which builds explainability into the model architecture, such as through modality attribution attention (e.g., ReasonNet), causal masking, or interpretable fusion gates. These methods offer stronger semantic alignment between decisions and feature sources but may introduce architectural complexity or inference overhead.

Recent works like ThinkTwice and ReasonNet have demonstrated the feasibility of integrating explainability mechanisms into fusion pipelines, enabling users to diagnose failure cases and uncover modality reliance under diverse driving scenarios. These approaches vary in their targeted explainability dimensions—modality-level, spatial-level, or temporal reasoning, and provide complementary insights for system auditing.

Some recent studies employ modality-level dropout or data masking during training to probe how much a model depends on each sensor. For example, Sensor Dropout, introduced in end-to-end multimodal sensor policy learning, randomly suppresses inputs from individual modalities (like camera, LiDAR, or GPS) during training. This encourages robustness to missing or corrupted inputs, enabling the network to maintain performance even when some sensors fail [84].

In mission-critical settings like agriculture or aviation, interpretable perception modules (e.g., rule-based confidence thresholds or redundancy checks) are often used as fail-safes. Autonomous driving could similarly benefit from hybrid fusion pipelines that integrate both learned and interpretable components—such as fusing deep outputs with traditional rule-based constraints.

Furthermore, the integration of uncertainty estimation and out-of-distribution detection is increasingly explored to enhance trustworthiness. Probabilistic fusion models like ADF [81] or epistemic uncertainty estimation via MC-Dropout can inform the confidence level of fused outputs, supporting decision-making modules in activating fallback strategies.

Another promising direction is causal analysis and counterfactual reasoning. For example, “what-if” explanations generated by altering one sensor’s input while keeping others fixed can help diagnose modality-specific failures. Research in causal interpretability for multi-modal perception is still nascent but holds potential for robust safety validation.

Finally, regulatory and certification frameworks are likely to mandate some level of interpretability in safety-critical perception systems. Developing real-time, human-understandable explanations for fusion-based predictions will be key to achieving public and regulatory trust in autonomous vehicles.

### 5.6. Data Scarcity and Annotation Cost

Building robust multi-sensor fusion models for autonomous driving critically depends on access to large-scale, high-quality, and precisely synchronized datasets annotated across modalities such as LiDAR, cameras, and Radar. However, acquiring such datasets is extremely expensive and labor-intensive. Annotating 3D bounding boxes for LiDAR or pixel-level semantic segmentation for images requires domain expertise, consistent calibration, and significant human resources. The challenge is further exacerbated in adverse weather conditions, nighttime scenarios, or occluded environments, where annotations become ambiguous and error-prone.

To alleviate this burden, synthetic data generation has been widely explored. Simulation platforms like CARLA, Synscapes, and Sim4CV-GTA provide cost-effective means to generate diverse and controllable environments with automatic labels. These systems simulate multiple sensor streams (e.g., LiDAR, camera, Radar) with perfect synchronization and produce ground-truth annotations for segmentation, depth, and detection. While they improve scalability and enable training on rare or dangerous scenarios, models trained solely on synthetic data often suffer from domain shifts when deployed in the real world due to sensor noise mismatch, visual artifacts, and unrealistic motion priors.

To bridge the domain gap, several learning strategies have been proposed:Self-supervised Pretraining: Models are pre-trained using proxy tasks such as masked reconstruction, contrastive matching, or temporal prediction. For instance, BEV-MAE [85] adopts masked autoencoding for BEV fusion pretraining, leveraging cross-view consistency without human labels.Pseudo-labeling and Label Propagation: Networks trained on labeled data are used to infer labels for unlabeled samples, which are then reused for retraining. For example, CRN [86] propagates spatial priors from Radar onto BEV maps, while L3PS [87] bootstraps LiDAR labels from camera segmentation masks.Cross-modal Consistency Learning: Multi-view contrastive losses or teacher-student networks enforce agreement between modalities. CM-KD [79] and CoMoFusion [88] employ knowledge distillation and contrastive embedding learning to align feature spaces across LiDAR and camera domains.Domain Adaptation and Style Transfer: Adversarial training or style translation methods reduce the distribution mismatch. For example, DA-MLF [89] performs adversarial alignment between real and simulated modalities, while CAPIT [90] applies image-to-image translation to adapt camera input to target domains.

Beyond algorithmic adaptation, data-efficient architectural design is gaining attention. Lightweight finetuning techniques like adapters or Low-Rank Adaptation (LoRA) enable model transfer with minimal labeled data. Such methods are particularly effective in vehicle-to-vehicle domain shift scenarios (e.g., from sedan to SUV) or regional transfer (e.g., Europe to Asia).

Cross-domain insights from other sectors also contribute valuable perspectives. In agricultural robotics, datasets often lack dense annotations due to vast fields and variable lighting. To overcome this, researchers apply weak supervision from GPS, topological maps, or seasonal crop models, as seen in [91]. Similarly, in autonomous driving, high-definition maps and V2X messages can provide coarse supervision signals [92,93], like static landmarks or occupancy priors, which help bootstrap sensor fusion labels.

Furthermore, multi-source weak supervision is emerging as a practical approach. For example, DriveLM [94] combines camera frames with Radar reflections and localization metadata to generate scene graphs, which are then refined through iterative reasoning. This reduces reliance on dense manual labels and improves robustness in edge cases like construction zones or ambiguous merges.

Nevertheless, even with improved learning techniques, the diversity of publicly available datasets remains limited. Most benchmarks, such as KITTI, nuScenes, Waymo [95] which focus on urban Western environments. Underrepresented domains such as rural roads, underpasses, developing countries, or extreme weather are rarely covered. As emphasized in [1], generalization across such domains remains a bottleneck for real-world deployment. Thus, there is an urgent need to build inclusive, globally representative datasets, with clearly defined sensor configurations and annotation standards.

In summary, overcoming data scarcity requires a multifaceted effort: leveraging simulation, exploiting unlabeled data, adapting across domains, and sharing diverse open datasets. Without addressing the annotation bottleneck, the advancement of robust and generalizable sensor fusion systems will remain constrained.

### 5.7. Evaluation Metrics and Benchmark Limitations

Current evaluation metrics and benchmark datasets for multi-sensor fusion systems such as KITTI, nuScenes, and Waymo Open Dataset, provide standard tasks like detection, tracking, and segmentation across various urban scenarios. However, these datasets lack scenarios with real-world degradations like adverse weather, partial sensor failure, or domain shifts caused by geographic variation. As a result, models that excel on these benchmarks often fail to generalize to diverse, unstructured, or safety-critical conditions encountered in deployment.

Moreover, the predominant metrics—mean Average Precision (mAP) for detection, Intersection-over-Union (IoU) for segmentation, and Average Displacement Error (ADE) for prediction—do not capture fusion-specific properties. They ignore critical aspects such as robustness to missing modalities, temporal synchronization issues, or fusion misalignment. For example, a model may achieve high mAP under clean input but degrade significantly with missing LiDAR frames, a vulnerability not reflected in the official scores.

To address these limitations, recent studies have proposed fusion-aware evaluation protocols. Modality ablation studies, where one or more sensors are dropped during inference, are used to assess the resilience of fusion models. For instance, M2BEV [4] evaluates performance under both RGB-only and LiDAR-only conditions to quantify fusion gain. Similarly, UniFusion [18] defines robustness curves that plot performance over increasing levels of simulated degradation, offering a more nuanced view of model stability.

Another direction is temporal robustness evaluation, where sensor streams are intentionally misaligned in time or corrupted with jitter to measure sensitivity. For example, TransFusion-L [32] incorporates temporal attention and explicitly quantifies the performance drop when timestamps are desynchronized.

Additionally, calibration perturbation testing has been employed in works like CalibNet [75] to study how small misalignments in extrinsic parameters affect the fusion pipeline. This highlights whether a model relies excessively on rigid geometric priors or can compensate through learned feature alignment.

Despite these efforts, no unified evaluation framework has been universally adopted. Most metrics remain task-specific and siloed, hindering apples-to-apples comparisons across fusion approaches. The lack of standardized degradation protocols also limits reproducibility and benchmarking transparency.

To improve future evaluations, a few directions are promising:Benchmark extensions with adverse conditions: Incorporating fog, rain, night-time driving, or partial sensor occlusion into datasets like nuScenes-rain or Waymo-weather would better reflect deployment scenarios.Multi-objective metrics: Proposals like Fusion Robustness Score (FRS) or Alignment-Weighted IoU (AW-IoU) aim to jointly measure accuracy, fusion consistency, and resilience to modality dropout.Task-specific fusion benchmarks: Dedicated benchmarks for fusion challenges—e.g., joint depth and semantics or cross-modality correspondence—would foster architectural innovation.

In summary, without standardized and fusion-aware metrics, model comparisons remain superficial. As fusion systems become central to safety-critical applications, the community must move toward holistic, degradation-resilient, and interpretable evaluation protocols that can ensure trustworthy performance in open-world environments.

### 5.8. Towards Foundation Models and Self-Adaptive Fusion

As multi-sensor fusion tasks grow in scale and complexity, there is a growing demand for general-purpose models that can adapt across diverse perception tasks and sensor configurations. Inspired by foundation models in natural language processing and vision, such as CLIP [96], DINO [97], and SAM [98] which recent work in autonomous driving has begun to explore large-scale pretraining across multiple modalities, aiming to unify semantic understanding and representation learning in a single framework.

Some fusion models attempt to build universal representations via multi-modal transformers, integrating inputs from cameras, LiDAR, and Radar into a shared BEV space. For example, BEVFusion proposes a unified BEV encoder that supports multiple downstream tasks like 3D object detection and semantic segmentation. Others, like M2BEV and MSAFusion [99], explore adaptive modality encoding to selectively fuse information based on task relevance and scene conditions.

However, scaling such models toward foundation-level capabilities presents critical challenges. First, real-time performance remains a bottleneck due to the computational cost of multi-modal fusion across high-resolution inputs. Second, interpretability becomes increasingly difficult as networks become deeper and more entangled, making it hard to analyze decision failures in safety-critical scenarios. Third, long-tail generalization—handling rare classes, edge-case dynamics, or novel weather conditions—requires continual adaptation beyond static offline training.

In response, there is a rising interest in self-adaptive fusion systems, which can dynamically assess sensor reliability and reconfigure the fusion strategy during runtime. For instance, AdaFusion [100] proposes attention-based gating mechanisms to downweight noisy or degraded modalities. Similarly, DGFEG [101] leverages a dynamic graph structure to route information adaptively between sensor-specific branches. These approaches aim to enhance robustness under partial sensor failure or domain shifts.

To achieve full self-adaptivity, future models must integrate principles from uncertainty modeling [102], meta-learning, and policy-driven fusion. Techniques such as dropout-based uncertainty estimation, test-time adaptation, and reinforcement-learned fusion policies could enable systems to select optimal modality combinations based on environmental context, reliability cues, or resource constraints.

Additionally, cross-task and cross-modal pretraining pipelines that support continual learning are essential to maintain performance as sensor suites evolve or new cities are deployed. For instance, multi-task supervision across detection, occupancy, and motion forecasting—as adopted in recent frameworks like Unified-IO [103] and Mask2Map [104] provide a scalable path toward shared representations that generalize across both tasks and sensors.

In conclusion, while multi-sensor fusion has traditionally been viewed as a static architecture problem, the rise of foundation models and adaptive learning marks a paradigm shift. Future systems must not only fuse modalities efficiently but also reason about fusion quality, adapt to degradation, and evolve with new data distributions—transforming static perception stacks into flexible, self-improving agents for real-world autonomy.

In conclusion, while multi-modal sensor fusion forms the backbone of modern autonomous driving perception systems, its practical deployment faces a multifaceted set of challenges. These span robustness, alignment, efficiency, generalization, and interpretability. Addressing these issues requires not only algorithmic innovation but also cross-domain inspiration, large-scale data infrastructure, and holistic system design thinking.

## 6. Future Directions and Research Prospects

With the emergence of next-generation deep learning models such as diffusion models, Mamba-based state space architectures, and large language models (LLMs), the landscape of multi-modal sensor fusion for autonomous driving is undergoing a paradigm shift. These techniques are not mere incremental improvements; they represent foundational advances that may reshape the design principles of fusion architectures, especially in perception and decision-making under complex, dynamic real-world environments. This chapter explores future directions from the perspective of how these advanced learning paradigms may address longstanding challenges such as robustness, generalization, explainability, and adaptability, thereby charting a new course for multi-modal fusion development.

To compare the performance of state-of-the-art multi-sensor fusion methods, Table 6 reports the mAP and NDS scores on the nuScenes test set, using the official six-camera and one-LiDAR configuration. All values are collected from original papers and evaluated under the official nuScenes metrics, without test-time augmentation or ensembling unless stated. While most methods focus on camera–LiDAR fusion, we include RCBEVDet, a representative camera–radar fusion method, to show the emerging trend of radar-based systems approaching LiDAR-level performance. However, the current best-performing solutions still rely on camera–LiDAR fusion for superior accuracy and robustness.

### 6.1. Structured Generation via Diffusion Models for Robust Fusion

Diffusion models, initially proposed for high-fidelity image generation [109,110], have demonstrated compelling capabilities in structured trajectory generation, scene completion, and sensor simulation. Their gradual denoising process aligns naturally with fusion settings that demand reconstruction from noisy, incomplete, or degraded inputs. The typical framework of diffusion-enhanced multi-modal fusion is shown in Figure 4.

In multi-sensor fusion, one emerging trend is the application of diffusion-based BEV reconstruction, where raw LiDAR and camera features are encoded into a latent space, and a diffusion decoder reconstructs complete BEV semantic or occupancy maps. Such approaches can inherently learn uncertainty-aware generation, enabling graceful degradation handling under sensor dropout. For instance, DiffBEV [111] and DifFUSER [106] have employed diffusion priors to recover occluded road topology and fine-grained semantics, outperforming deterministic fusion counterparts.

Furthermore, diffusion can be employed to simulate missing modalities during training, thereby enhancing robustness to partial observability. By modeling conditional distributions of one sensor modality given another (e.g., p(LiDAR∣camera)), these models allow for more resilient fusion under failure conditions.

Future work should explore conditional diffusion models in which navigation priors (e.g., HD maps, goal points) or dynamic contexts (e.g., traffic light state) guide fusion output generation. Integration with uncertainty quantification tools could further support safety-critical perception.

### 6.2. Long-Context Fusion with Mamba and State Space Models

Transformer-based fusion models like TransFuser [2] have achieved remarkable performance in reasoning over multi-modal sensor data. However, their computational cost and inefficiency in long-sequence modeling limit their deployment in real-time driving settings. State space models, notably Mamba [112], have recently emerged as lightweight yet expressive alternatives, the framework that fusion with Mamba is shown in Figure 5.

Mamba’s key advantage lies in its linear-time sequence processing and capability to model structured recurrence. This allows multi-modal fusion to benefit from long-horizon temporal memory without incurring transformer-like overhead. When applied to sensor fusion, Mamba blocks can replace attention layers to encode cross-modal temporal dynamics (e.g., delayed Radar returns, asynchronous LiDAR scans) more efficiently.

Recent studies [113] demonstrate that Mamba-based models can maintain temporal consistency in motion forecasting, 3D object tracking, and even behavior prediction. This motivates their integration into fusion perception pipelines, especially when modeling temporal alignment, latency compensation, or vehicle ego-motion priors.

Future directions include combining Mamba with diffusion-based modules, which use Mamba to model temporal priors and diffusion to handle spatial uncertainty—to enable spatio-temporal generative fusion. Additionally, fusing Mamba with continuous memory architectures could address online adaptation and continual learning.

### 6.3. Semantic Reasoning and Fusion Guidance with LLMs

LLMs particularly vision–language–action (VLA) frameworks, are gaining traction in autonomous driving for their ability to bridge semantic understanding with decision-making. While most fusion methods rely purely on perception signals, LLMs offer contextual reasoning that can guide multi-modal alignment and fusion decisions.

For example, systems like VLA-Drive [114] and Prompt4AD [115] integrate LLMs with BEV features to provide goal-driven planning or linguistic command execution. These models can be extended to fusion strategy modulation: based on the semantic context (e.g., “rainy night at intersection”), the LLM selects sensor weights, highlights key features, or even queries external map priors.

Moreover, LLMs can serve as fusion outcome explainers, translating raw sensor disagreements into human-understandable feedback (e.g., “LiDAR missing due to heavy rain; fallback to Radar depth”). Paired with XAI modules [116], LLMs could assist in model auditing, error localization, and post-deployment monitoring.

A promising avenue is to train LLMs not just as planners, but as fusion orchestrators, dynamically assembling fusion submodules based on the current task, reliability score, or mission context. Challenges remain in ensuring real-time latency and grounded predictions, but hybrid token-level fusion with pre-trained LLMs is an active research frontier.

### 6.4. Toward Universal Fusion Foundation Models

Inspired by CLIP, SAM, and Flamingo [117] in vision–language domains, the community is exploring foundation models for sensor fusion, large-scale pre-trained backbones trained across multiple modalities, tasks, and environments. Unlike task-specific fusion architectures, these models aim to learn general-purpose representations that transfer across datasets and geographies.

Early attempts like TransFusion, BEVFusion, and CMT pretrain encoders on multi-modal perception (e.g., camera + LiDAR), but remain constrained to fixed sensor types. A true foundation fusion model would:Accept arbitrary sensor combinations as input (plug-and-play).Output universal representations (e.g., BEV maps, occupancy fields, trajectories).Transfer across domains with minimal fine-tuning.

This direction demands massive multi-modal datasets with high annotation diversity. Self-supervised training with cross-modal contrastive loss, masked reconstruction, and diffusion pretext tasks can bootstrap training at scale. Moreover, multi-modal alignment tokens, similar to vision–language alignment in BLIP [118], can guide interaction between modalities.

By combining the structured reasoning of LLMs, generative fidelity of diffusion, and temporal expressivity of Mamba, foundation fusion models can become both generalizable and controllable. A modular backbone (e.g., a camera encoder, a LiDAR encoder, a fusion head, and a planner head) may soon dominate real-world autonomous driving stacks.

### 6.5. Self-Adaptive and Continually Evolving Fusion Systems

Static fusion models, once deployed, often suffer performance degradation under changing environments, sensor wear-out, or unseen scenes. To address this, self-adaptive fusion systems must evolve beyond fixed pipelines to support lifelong learning, policy-driven configuration, and dynamic recovery [99].

Core enablers include:Online uncertainty estimation to modulate fusion weights [119].Memory-augmented architectures (e.g., retrieval-based Mamba) for context-aware fusion.Reinforcement learning (RL) policies to optimize sensor selection or fusion mode based on downstream performance (e.g., braking accuracy, latency) [120].Curriculum learning frameworks to gradually expose fusion models to harder conditions (e.g., sensor failure, low light) [121].

Furthermore, continual adaptation requires stability-plasticity balancing: ensuring models retain prior skills while adapting to new ones. Tools like Elastic Weight Consolidation [122], replay buffers, or generative rehearsal (e.g., using diffusion to simulate past scenes) may assist in this.

Finally, self-adaptive fusion systems must be interpretable, fail-safe, and certifiable. Coupling real-time causal attribution, uncertainty thresholds, and fallback rules (e.g., logic override under low sensor confidence) is essential for deployment-grade trust.

In summary, the future of multi-modal sensor fusion lies in a cross-pollination of deep learning advances: generative modeling (diffusion), efficient sequence modeling (Mamba), and contextual reasoning (LLMs). These paradigms promise not only to enhance fusion accuracy and robustness but also to enable transparent, adaptable, and scalable autonomous systems. The fusion architectures of tomorrow will likely be modular, foundation-based, and self-evolving, which will usher in a new era of intelligence at the intersection of perception, semantics, and control.

## 7. Conclusions

We present a comprehensive review of recent progress in multi-modal sensor fusion for autonomous driving, spanning from fusion architectures and task-specific adaptations to practical deployment challenges. While significant improvements have been made, especially through BEV representations, attention-based alignment, and modality-specific supervision, real-world deployment still faces hurdles in generalization, interpretability, and resilience. Emerging paradigms, such as generative modeling via diffusion, efficient sequential reasoning using Mamba-like networks, and the interpretive power of large language models, offer promising directions for advancing fusion beyond task performance. Future efforts must emphasize robustness to modality failure, domain shift adaptation, and human-centric trust to enable safe and certifiable fusion-driven autonomy.

## Figures and Tables

**Figure 1 sensors-25-06033-f001:**
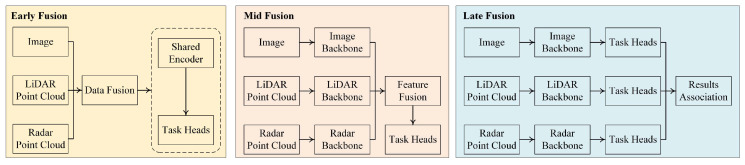
Different multi-sensor fusion process.

**Figure 2 sensors-25-06033-f002:**
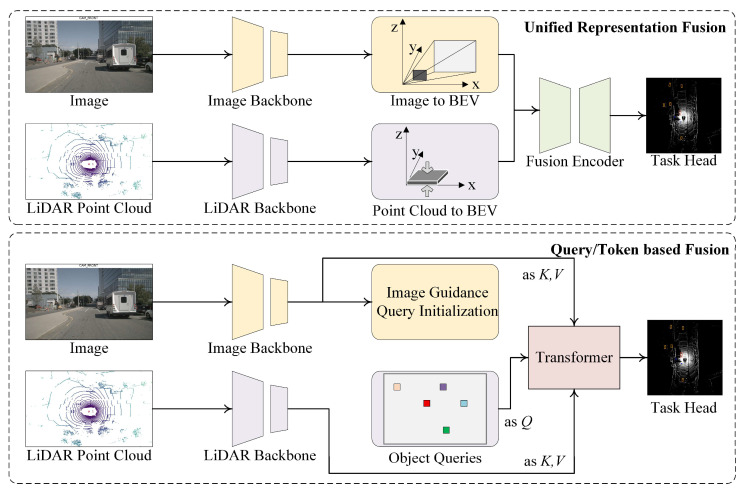
Mainstream Camera–LiDAR fusion architectures.

**Figure 3 sensors-25-06033-f003:**
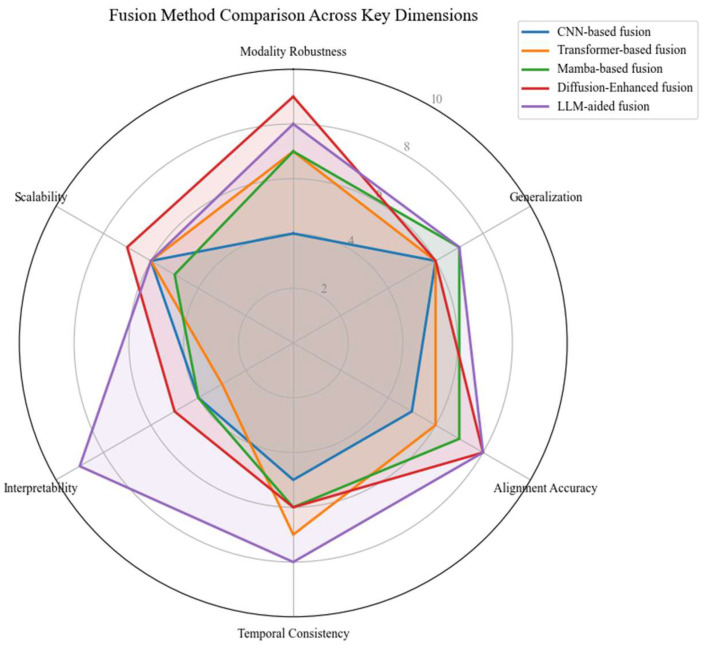
Fusion method comparison across key dimensions. Radar chart comparison of five fusion paradigms across six key evaluation dimensions: modality robustness, generalization, alignment accuracy, temporal consistency, interpretability, and scalability. CNN-based fusion shows limited flexibility and robustness under dynamic conditions. Transformer-based and Mamba-based methods enhance alignment and sequence modeling, but may lack interpretability. Diffusion-enhanced fusion achieves strong structural alignment and robustness, while LLM-aided fusion exhibits high interpretability and adaptability across modalities. The chart highlights the trade-offs and complementarities among modern fusion strategies. The scores in this radar chart are trend-based qualitative estimates derived from architecture analysis and comparative studies in existing literature, not from a single experimental benchmark.

**Figure 4 sensors-25-06033-f004:**
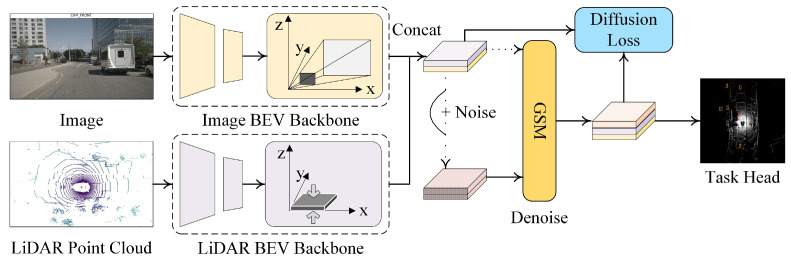
Overview of a diffusion-enhanced multi-modal fusion framework. Image and LiDAR inputs are first encoded into BEV feature maps via lightweight modality-specific backbones. The fused representation is then perturbed with noise and passed through a Gaussian denoising module (GSM), trained using a diffusion loss. The denoised output is fed into a task head for downstream objectives such as 3D detection or segmentation. This architecture enables structure-aware fusion and robust representation reconstruction.

**Figure 5 sensors-25-06033-f005:**
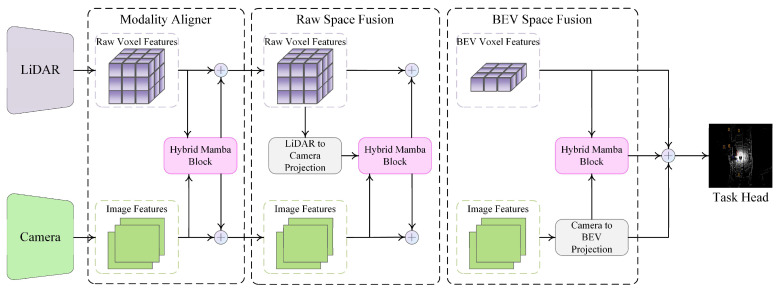
Architecture of a hierarchical Mamba-based fusion framework. The pipeline integrates LiDAR and camera features across three stages: (1) Modality Aligner aligns raw modality-specific voxel and image features through a Hybrid Mamba Block; (2) Raw Space Fusion projects LiDAR features into the camera view for cross-domain reasoning; (3) BEV Space Fusion consolidates fused representations into a unified BEV space. Each stage leverages Hybrid Mamba Blocks for temporal-aware feature refinement and structured alignment, culminating in task-specific prediction through the final head.

**Table 1 sensors-25-06033-t001:** Comparative Analysis of Theoretical Foundations in Sensor Fusion.

Approach	Reasoning Paradigm	Prior Knowledge	Fusion Stage	Strengths
Bayesian Filtering	Probabilistic Inference	Strong priors	Early/Middle	Handles uncertainty, recursive updates
Multi-View Learning	Representation Alignment	Weak priors	Late Fusion	Learns from view diversity
Uncertainty Modeling	Variational Approximation	Moderate priors	Middle/Late	Enhances robustness, supports risk-awareness
Transformer-based Fusion	Attention-based Structure	No explicit prior	All Stages	Captures global dependencies

**Table 2 sensors-25-06033-t002:** Characteristics of different modalities sensors.

Type	Advantages	Limitations
Camera	Rich semantic information, affordable hardware, passive sensing.	Sensitive to lighting conditions (e.g., night, glare), weather (e.g., fog, rain), and lack depth information without additional geometry.
LiDAR	Accurate depth perception, strong spatial resolution, invariant to lighting.	High cost, mechanical complexity (for spinning LiDAR), reduced performance in fog or rain.
Radar	Robust to lighting and weather, direct velocity measurement.	Low spatial resolution, noisy data, ghost objects.
Ultrasonic sensor	Low cost, simple integration.	Limited range (~2–5 m), poor angular resolution, unreliable in high-speed scenarios.
IMUs	High-frequency data, unaffected by external conditions.	Prone to drift, requires integration with other sensors for accurate long-term localization.
GNSS	Global positioning, widely available.	Prone to multipath errors, signal loss in urban canyons or tunnels.

**Table 3 sensors-25-06033-t003:** Comparative Analysis of Fusion Methodologies.

Fusion Methodology	Structural Assumptions	Data Dependency	Adaptability	Advantages
Complementarity in Fusion	Assumes informative diversity	Moderate	Task-specific	Exploits heterogeneous sensor strengths
Probabilistic Fusion Models	Likelihood-based reasoning	High	Static environments	Handles noise and ambiguity
Hybrid & Self-Supervised Approaches	Multi-loss or multi-branch	Low-to-moderate	Generalizable	Reduces labeling cost, learns semantics
Online Adaptation & Continual Learning	Temporal shift awareness	Dynamic data	High adaptability	Supports evolving environments

**Table 4 sensors-25-06033-t004:** Characteristics of Representative Multi-Modal Autonomous Driving Datasets.

Dataset	Modalities	Tasks Supported	Weather & Time Diversity	Annotation Quality
KITTI	Camera, LiDAR, GPS, IMU	Detection, Tracking, Depth	Limited (Daylight only)	Moderate (Sparse LiDAR, 2D/3D Boxes)
nuScenes	Camera (6), LiDAR, Radar, GPS, IMU	Detection, Tracking, Segmentation	High (Night, Rain, Fog)	Rich (360°, 3D, 2 Hz)
Waymo	Camera (5), LiDAR (5)	Detection, Tracking	Medium (Day/Night, Light Rain)	High (Dense LiDAR, HD Maps)
PandaSet	Camera, LiDAR, Radar	Detection, Segmentation	Medium (Clear to Rain)	Detailed (Point-level Labels)
RADIATE	Camera, LiDAR, Radar, GPS, IMU	Detection, Tracking, Weather Testing	Very High (Snow, Fog, Rain, Night)	Dense (Multi-weather frames)

**Table 5 sensors-25-06033-t005:** Task-Specific Challenges and Representative Fusion Methods.

Task Type	Key Fusion Challenges	Representative Methods
Depth Completion	Sparse-to-dense LiDAR reconstruction, geometric projection error, uncertainty modeling	DeepLiDAR, FusionNet, TransDepth
Dynamic Object Detection	Temporal misalignment, scale variance, occlusion handling, motion-aware fusion	TransFusion, ContFuse, BEVFusion
Static Object Detection	Long-range low-SNR targets, geometric detail preservation, semantic boundary ambiguity	CenterFusion, HVNet, FusionPainting
Semantic & Instance Segmentation	Cross-modal semantic alignment, class imbalance, fine-grained spatial matching	3DMV, MVPNet, DeepInteraction
Multi-Object Tracking	Temporal consistency, identity preservation, modality-dependent Re-ID drift	BEVTrack, VPFNet, Sparse4D
Online Cross-Sensor Calibration	Real-time extrinsic drift compensation, cross-modal geometric alignment	CalibNet, RegNet, DeepCalib

**Table 6 sensors-25-06033-t006:** Comparison of 3D object detection performance on the nuScenes test dataset. Comparison of recent multi-modal 3D object detection methods evaluated on the nuScenes test set. All methods use camera and LiDAR inputs except RCBEVDet. Metrics include mean Average Precision (mAP) and nuScenes Detection Score (NDS), where higher values indicate better performance. The table highlights a progressive performance improvement, with MambaFusion achieving state-of-the-art results, reflecting the efficacy of lightweight sequential modeling in multi-sensor fusion. **↑** indicates that higher values are better.

Method	Modality	Reference	mAP ↑	NDS ↑
TransFusion	Camera + LiDAR	CVPR 2022	68.9	71.6
DeepInteraction	Camera + LiDAR	NeurIPS 2022	70.8	73.4
BEVFusion	Camera + LiDAR	ICRA 2023	70.2	72.9
MSMDFusion [105]	Camera + LiDAR	CVPR 2023	71.0	73.0
CMT	Camera + LiDAR	ICCV 2023	72.0	74.1
DifFUSER [106]	Camera + LiDAR	ECCV 2024	71.3	73.8
RCBEVDet [107]	Camera + Radar	CVPR 2024	67.3	72.7
IS-FUSION [108]	Camera + LiDAR	CVPR 2024	73.0	75.2
MambaFusion	Camera + LiDAR	ICCV 2025	73.2	75.9

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
