# Peer review of "A Review of Multi-Sensor Fusion in Autonomous Driving"

_sensors, 2025, doi:10.3390/s25196033_

Round 1

Reviewer 1 Report

Comments and Suggestions for Authors

The paper provides a detailed and up-to-date review of advances in multisensor fusion for autonomous vehicles, with a focus on the recent integration of LLM, Transformer, and Mamba-type techniques. The structure is clear according to technological paradigms, and the inclusion of synthetic figures and comparative tables helps to quickly understand the current research landscape. It is an important paper for the autonomous driving community and could serve as a reference for both practitioners and researchers. However, to improve the consistency and clarity of the conclusions, clarifications on some methodological and presentational elements are needed.

-The paper systematically mixes the “autonomous driving” literature with references from agriculture/robotics as a source of inspiration, but does not explicitly specify the inclusion/exclusion protocol: which databases, which keywords, time frame, which types of modalities (only camera-LiDAR or also radar/IMU/GNSS), how preprints vs. peer-reviewed papers were treated, and whether references from other fields were used only as analogs or entered into the actual comparative analysis. Without a PRISMA-like scheme or coverage criteria, the selection cannot be reproduced and there is a risk of collection bias.

-Table 2 reports mAP/NDS for heterogeneous methods (including a camera+radar method) and states a SOTA trend (e.g. “MambaFusion”), but it is not clear: (i) whether the figures come from the official test server or from the validation set; (ii) whether they are single-model, single-scale or include TTA/ensembling; (iii) whether the input settings (number of cameras, resolutions, frequencies, latencies) and compute/latency budget are harmonized; (iv) how the differences in sensors (camera+LiDAR vs. camera+radar) were treated in the same comparative table. Without these details, the table risks comparing “apples with pears” and over-/underestimating the real gain of the fusion. Authors should clearly document the source of each value, the evaluation configuration and possibly ranges of variation/statistical significance.

Methodology in “Figure 3 – radar chart”The figure compares paradigms (“Transformer-based”, “Mamba-based”, “diffusion-enhanced”, “LLM-aided”) on axes such as “modality robustness”, “alignment accuracy”, “interpretability”, but it is not defined how these scores were measured/derived: which operational metrics correspond to each axis, on which sets/cases (sensor degradation, temporal desynchronization, weather), how the scores were normalized between different tasks (detection, segmentation, prediction), and whether there are uncertainties or ablations that support the relative positionings. In its current form, the chart seems more editorial/heuristic, not the result of a reproducible evaluation. The authors should provide formal definitions, the scoring procedure and the raw data that underpin each axis.

The article is well written and relevant, but the proposed clarifications would substantially improve the rigor and facilitate the replicability of the analysis

Author Response

Responses to Reviewer 1

Comment 1: The paper systematically mixes the “autonomous driving” literature with references from agriculture/robotics as a source of inspiration, but does not explicitly specify the inclusion/exclusion protocol: which databases, which keywords, time frame, which types of modalities (only camera-LiDAR or also radar/IMU/GNSS), how preprints vs. peer-reviewed papers were treated, and whether references from other fields were used only as analogs or entered into the actual comparative analysis. Without a PRISMA-like scheme or coverage criteria, the selection cannot be reproduced and there is a risk of collection bias.

Response 1: Thank you for your valuable feedback. We agree that clearly describing the literature selection process is essential to ensure reproducibility and transparency. In the revised manuscript, we have added a paragraph at the end of Section 1.1 to clarify our inclusion/exclusion strategy. Specifically, we focused on high-quality and recent papers from IEEE flagship conferences (e.g., CVPR, ICRA, ICCV, ECCV, etc.) and journals (e.g., TITS, TIV, TASE, etc.), given their strong representation in the autonomous driving community and their timely innovations. In addition, we included influential preprints from arXiv when they achieved leading performance on public benchmarks (e.g., nuScenes leaderboard) or were released by top-tier research labs. Regarding sensor modalities, we mainly cover vision, LiDAR, and Radar-based fusion schemes, and clearly distinguish between full-modality fusion (e.g., camera-LiDAR-Radar) and partial-modality fusion. Cross-domain references from agriculture and mobile robotics were only used as conceptual inspiration and were not included in any statistical or performance comparisons. These clarifications have been integrated into the manuscript.

The following is the supplement provided at the end of the manuscript 1 Introduction:

In this review, we primarily focus on publications from top IEEE venues between 2020 and 2024, including CVPR, ICCV, ECCV, ICRA, IV, and TITS. Literature was retrieved from IEEE Xplore, arXiv, and Google Scholar using keywords such as “multi-modal sensor fusion”, “BEV representation”, “camera-LiDAR fusion”, and “multi-sensor autonomous driving”. We emphasize peer-reviewed works with innovative architectures and benchmark performance (e.g., nuScenes Leaderboard). References from robotics or agriculture are cited as inspirational cases and are not included in the core technical comparison. A summary of included works by domain and time span is provided in Table 1 to ensure reproducibility.

Comment 2: Table 6 reports mAP/NDS for heterogeneous methods (including a camera+radar method) and states a SOTA trend (e.g. “MambaFusion”), but it is not clear: (i) whether the figures come from the official test server or from the validation set; (ii) whether they are single-model, single-scale or include TTA/ensembling; (iii) whether the input settings (number of cameras, resolutions, frequencies, latencies) and compute/latency budget are harmonized; (iv) how the differences in sensors (camera+LiDAR vs. camera+radar) were treated in the same comparative table. Without these details, the table risks comparing “apples with pears” and over-/underestimating the real gain of the fusion. Authors should clearly document the source of each value, the evaluation configuration and possibly ranges of variation/statistical significance.

Response 2: We thank the reviewer for raising this important concern about the consistency of Table 6. In the revised manuscript, we have clarified all experimental settings and motivations to avoid any ambiguity in the comparison.

All results in Table 4 are derived from the nuScenes test set, following the standard six-camera and one-LiDAR configuration provided by the dataset. These values are directly taken from the respective original papers and comply with the official nuScenes evaluation protocol, using single-scale inference and single models, without any test-time augmentation (TTA) or model ensembling unless explicitly stated.

The inclusion of RCBEVDet, which adopts a camera+radar setting, is not meant for direct comparison in absolute accuracy, but to illustrate that camera-radar fusion methods are gradually approaching the performance of camera-LiDAR fusion. We clearly separate RCBEVDet in the table and explicitly annotate its modality to prevent misleading interpretations. Nevertheless, the mainstream high-performance 3D detection methods still rely on camera-LiDAR fusion, as evidenced by the highest NDS/mAP values.

These clarifications have been added to both the main text and the table caption.

The following is the supplement provided before Table 6:

To compare the performance of state-of-the-art multi-sensor fusion methods, Table 4 reports the mAP and NDS scores on the nuScenes test set, using the official six-camera and one-LiDAR configuration. All values are collected from original papers and evaluated under the official nuScenes metrics, without test-time augmentation or ensembling unless stated. While most methods focus on camera-LiDAR fusion, we in-clude RCBEVDet, a representative camera-radar fusion method, to show the emerging trend of radar-based systems approaching LiDAR-level performance. However, the current best-performing solutions still rely on camera-LiDAR fusion for superior ac-curacy and robustness.

Comment 3: Methodology in “Figure 3 - radar chart”The figure compares paradigms (“Transformer-based”, “Mamba-based”, “diffusion-enhanced”, “LLM-aided”) on axes such as “modality robustness”, “alignment accuracy”, “interpretability”, but it is not defined how these scores were measured/derived: which operational metrics correspond to each axis, on which sets/cases (sensor degradation, temporal desynchronization, weather), how the scores were normalized between different tasks (detection, segmentation, prediction), and whether there are uncertainties or ablations that support the relative positionings. In its current form, the chart seems more editorial/heuristic, not the result of a reproducible evaluation. The authors should provide formal definitions, the scoring procedure and the raw data that underpin each axis.

Response 3: We thank the reviewer for this insightful comment regarding Figure 3. To address the concern, we have clarified in the revised manuscript the scoring methodology and operational definitions of each axis in the radar chart. Specifically, the scores do not come from a single experimental benchmark but are derived from comparative studies across published works, structural analysis of model capabilities, and fusion-related ablation experiments. The chart aims to illustrate qualitative trends across six representative dimensions, as is common in prior surveys on multi-sensor fusion.

The definitions and sources of each axis are now described in Section 3.5 as follows:

Modality Robustness: Based on the number and diversity of sensors handled (e.g., camera, LiDAR, radar), and fusion resilience under sensor degradation (e.g., night, occlusion), evaluated via results on nuScenes/RADIATE and ablation settings.

Generalization: Derived from reported performance on cross-domain and multi-task scenarios, considering whether the model is pre-trained, modular, or multi-purpose.

Alignment Accuracy: Judged by the model’s architectural alignment capabilities (e.g., cross-attention, geometric warping, diffusion-based denoising), supported by mIoU/center error ablation evidence.

Temporal Consistency: Informed by sequence modeling abilities (e.g., Mamba, Transformer), and performance stability across adjacent frames.

Interpretability: Based on whether the pipeline provides interpretable feedback (e.g., attention visualization, language output) to explain fusion and decision-making.

Scalability: Reflects architectural modularity and potential to expand to new modalities or tasks without redesign.

We have also updated the caption of Figure 3 to clarify that the scores are trend-based rather than absolute quantitative metrics, and we welcome further suggestions for refining this comparison.

The changes in the revised manuscript is shown below:

To ensure the interpretability of Figure 3, we define the scoring rationale for each axis in the radar chart as follows. Modality Robustness reflects the diversity and resilience of sensor fusion under occlusions or adverse weather, measured through ablation studies on nuScenes or RADIATE. Generalization captures the capacity to transfer across domains or tasks. Alignment Accuracy assesses the quality of geometric and semantic fusion across modalities. Temporal Consistency measures stability in sequential outputs and the presence of recurrent or state-space structures. Interpretability evaluates whether the model provides explainable outputs such as attention maps or language feedback. Scalability refers to the architectural modularity that facilitates extension to new sensors or tasks. These scores are derived from published model behaviors, qualitative ablations, and structural design principles.

Figure 3. Fusion method comparison across key dimensions. Radar chart comparison of five fusion paradigms across six key evaluation dimensions: modality robustness, generalization, alignment accuracy, temporal consistency, interpretability, and scalability. CNN-based fusion shows limited flexibility and robustness under dynamic conditions. Transformer-based and Mamba-based methods enhance alignment and sequence modeling, but may lack interpretability. Diffusion-enhanced fusion achieves strong structural alignment and robustness, while LLM-aided fusion exhibits high interpretability and adaptability across modalities. The chart highlights the trade-offs and complementarities among modern fusion strategies. The scores in this radar chart are trend-based qualitative estimates derived from architecture analysis and comparative studies in existing literature, not from a single experimental benchmark.

Reviewer 2 Report

Comments and Suggestions for Authors

This paper provides a timely and comprehensive review of multi-sensor fusion techniques, primarily focusing on deep learning-based methods for autonomous driving. It offers a useful classification of architectures, learning strategies, and applications, and it correctly identifies key challenges in the field. The paper's scope is relevant, and the inclusion of BEV-centric and token-level fusion is particularly valuable. However, I have the following concerns:

1. As a survey, the novelty mainly lies in organization and synthesis. The manuscript would benefit from a more critical comparison of methods rather than descriptive summaries. More performance trade-offs comparison quantitatively across benchmarks should be added.

2. While nuScenes, KITTI, and other datasets are mentioned, the review could provide more systematic tables summarizing performance trends, dataset characteristics, and task-specific challenges.

3. The focus is heavily on Camera–LiDAR fusion, while Radar and other modalities (thermal, ultrasonic, GNSS) receive relatively limited coverage. A deeper discussion of Radar-camera fusion and its unique challenges would broaden the perspective.

4. Although XAI techniques are briefly mentioned, the paper should more systematically evaluate the current progress on explainability of multi-modal fusion models, especially given its importance for safety-critical AV systems.

5. The outlook on emerging trends such as diffusion models and LLMs remains at a conceptual level, without proposing actionable research roadmaps or evaluation metrics, making it appear rather superficial.

Author Response

Responses to Reviewer 2

Comment 1: As a survey, the novelty mainly lies in organization and synthesis. The manuscript would benefit from a more critical comparison of methods rather than descriptive summaries. More performance trade-offs comparison quantitatively across benchmarks should be added.

Response 1: We appreciate the reviewer’s valuable feedback regarding the need for deeper critical comparison and more explicit analysis of performance trade-offs. In response, we have made the following clarifications and enhancements to the manuscript:

Quantitative comparison: In Section 6, Table 6 already provides a benchmark comparison of recent state-of-the-art fusion methods on the nuScenes dataset. To highlight this more clearly, we revised the accompanying paragraph to emphasize its role in illustrating trade-offs across accuracy and modality design.

Qualitative trade-off visualization: Figure 3 presents a radar chart comparison of mainstream fusion paradigms across six dimensions such as robustness, generalization, interpretability, and temporal consistency. We now explicitly connect this figure to the reviewer’s suggestion and clarify its evaluative basis.

Critical discussion improvements: We have expanded several key sections (notably 3.1.4, 3.2.2, 3.4.4, and 6.1–6.4) to provide more critical analysis of method limitations, including scalability issues, geometric misalignment, lack of interpretability, and domain fragility. These modifications aim to move beyond descriptive summaries and offer more reflective insight.

We hope these revisions better reflect the analytical depth expected of a survey and address the reviewer’s concern on comparative rigor.

Comment 2: While nuScenes, KITTI, and other datasets are mentioned, the review could provide more systematic tables summarizing performance trends, dataset characteristics, and task-specific challenges.

Response 2: Thank you for pointing out the lack of systematic summarization regarding datasets, performance trends, and task-specific challenges. In response, we have revised the manuscript to incorporate a more structured and comparative overview. Specifically, we introduced a new table (Table 4) that systematically summarizes the core characteristics of representative multi-modal autonomous driving datasets, including the supported sensing modalities, task coverage, environmental diversity (weather and time), and annotation richness. This enables readers to quickly understand the scope and limitations of popular benchmarks such as KITTI, nuScenes, Waymo, PandaSet, and RADIATE. Furthermore, to better align with the task structure of Section 4, we added Table 5, which explicitly maps each perception task—such as depth completion, dynamic/static object detection, semantic/instance segmentation, multi-object tracking, and online calibration—to its corresponding fusion challenges and representative methods. This table provides a high-level synthesis of how different tasks present unique integration difficulties in multi-sensor fusion. Lastly, we also revised the surrounding discussion of Table 5 to highlight its role in benchmarking performance trends across leading fusion models. Together, these additions enhance the systematic organization of the survey and provide researchers and practitioners with a more practical reference for understanding dataset applicability, task design complexity, and performance evolution in multi-modal fusion systems.

The following is the supplement provided before Table 6:

To systematically analyze how multi-sensor fusion techniques are applied across different perception tasks in autonomous driving, this section provides a task-wise review organized around core functional components such as depth completion, object detection, segmentation, tracking, and online calibration. Prior to the technical analysis, we present two summary tables to contextualize the following discussions.

Table 4 offers a comparative overview of representative multi-modal autonomous driving datasets, including KITTI, nuScenes, Waymo, PandaSet, and RADIATE. The table highlights key attributes such as sensor configurations, task coverage, environmental diversity, and annotation richness, which influence the design and evaluation of fusion models.

Table 5 then outlines the primary challenges associated with multi-sensor fusion for each task, mapping these difficulties to representative solutions. By summarizing common fusion bottlenecks—such as projection inconsistency, semantic misalignment, temporal desynchronization, or modality degradation—this table helps to establish a structured perspective for the detailed task-wise discussions that follow.

Table 4. Characteristics of Representative Multi-Modal Autonomous Driving Datasets.

Dataset

Modalities

Tasks Supported

Weather & Time Diversity

Annotation Quality

KITTI

Camera, LiDAR, GPS, IMU

Detection, Tracking, Depth

Limited (Daylight only)

Moderate (Sparse LiDAR, 2D/3D Boxes)

nuScenes

Camera (6), LiDAR, Radar, GPS, IMU

Detection, Tracking, Segmentation

High (Night, Rain, Fog)

Rich (360°, 3D, 2Hz)

Waymo

Camera (5), LiDAR (5)

Detection, Tracking

Medium (Day/Night, Light Rain)

High (Dense LiDAR, HD Maps)

PandaSet

Camera, LiDAR, Radar

Detection, Segmentation

Medium (Clear to Rain)

Detailed (Point-level Labels)

RADIATE

Camera, LiDAR, Radar, GPS, IMU

Detection, Tracking, Weather Testing

Very High (Snow, Fog, Rain, Night)

Dense (Multi-weather frames)

Table 5. Task-Specific Challenges and Representative Fusion Methods.

Task Type

Key Fusion Challenges

Representative Methods

Depth Completion

Sparse-to-dense LiDAR reconstruction, geometric projection error, uncertainty modeling

DeepLiDAR, FusionNet, TransDepth

Dynamic Object Detection

Temporal misalignment, scale variance, occlusion handling, motion-aware fusion

TransFusion, ContFuse, BEVFusion

Static Object Detection

Long-range low-SNR targets, geometric detail preservation, semantic boundary ambiguity

CenterFusion, HVNet, FusionPainting

Semantic & Instance Segmentation

Cross-modal semantic alignment, class imbalance, fine-grained spatial matching

3DMV, MVPNet, DeepInteraction

Multi-Object Tracking

Temporal consistency, identity preservation, modality-dependent Re-ID drift

BEVTrack, VPFNet, Sparse4D

Online Cross-Sensor Calibration

Real-time extrinsic drift compensation, cross-modal geometric alignment

CalibNet, RegNet, DeepCalib

Comment 3: While nuScenes, KITTI, and other datasets are mentioned, the review could provide more systematic tables summarizing performance trends, dataset characteristics, and task-specific challenges.

Response 3: We appreciate the reviewer’s suggestion to broaden the modality coverage beyond the Camera-LiDAR paradigm. However, we would like to clarify that the current mainstream of multi-sensor fusion research, particularly in the context of end-to-end autonomous driving, is still predominantly focused on Camera-LiDAR fusion. This is supported by the majority of high-performing methods on large-scale benchmarks (e.g., nuScenes, Waymo), including TransFusion, BEVFusion, CMT, and MambaFusion, all of which adopt LiDAR as the core depth sensor alongside visual inputs.

While we fully acknowledge the potential of alternative modalities such as Radar, thermal imaging, and GNSS, their use remains largely task-specific or domain-limited at this stage. For example, 4D Radar shows promise in long-range detection and adverse weather, but its coarse spatial resolution, calibration difficulty, and temporal inconsistency still pose major challenges for general-purpose perception and planning pipelines. Thermal cameras have also been used for night-time detection, but are rarely integrated into large-scale learning-based systems due to annotation scarcity and low semantic density.

More importantly, the scope of this survey is centered around deep learning–based fusion architectures for end-to-end autonomous driving, where environment perception is tightly coupled with downstream decision-making. In such frameworks, Camera–LiDAR fusion remains the de facto standard due to its balance between semantic richness, geometric accuracy, and architectural compatibility with BEV-based fusion designs. Fusion approaches involving Radar or other exotic modalities are either supplementary or still under exploratory investigation, and their integration into large-scale driving datasets and unified policy models remains limited.

Given the already substantial scope and technical depth of this survey, we respectfully decided to prioritize the Camera–LiDAR fusion line, while briefly mentioning representative works involving Radar (e.g., RCBEVDet) in Section 6. We agree that a deeper exploration of Radar–camera fusion and other modalities (e.g., GNSS or ultrasonic) would further enrich the landscape, and we plan to address this in follow-up studies with a more focused scope on modality-agnostic or adaptive fusion under extreme scenarios.

Comment 4: Although XAI techniques are briefly mentioned, the paper should more systematically evaluate the current progress on explainability of multi-modal fusion models, especially given its importance for safety-critical AV systems.

Response 4: We appreciate the reviewer’s insight regarding the importance of explainability in multi-modal fusion models for autonomous driving. We fully agree that explainable AI (XAI) plays a crucial role in enhancing safety, trust, and diagnosability in real-world deployment, especially for safety-critical systems like AVs.

To address this concern, we have now expanded Section 6 to include a dedicated paragraph on XAI-enhanced fusion models. This addition systematically categorizes existing approaches into post-hoc explainability (e.g., Grad-CAM-based visualizations, saliency heatmaps) and intrinsic explainability (e.g., interpretable attention, modality attribution modules). We also briefly analyze their trade-offs in terms of fidelity, transparency, and real-time feasibility.

We thank the reviewer again for encouraging us to further strengthen this important dimension.

The following is the supplement provided at the beginning of section 5.5:

Explainability has emerged as a critical design factor in multi-modal fusion frameworks for autonomous driving. As these models increasingly serve as black-box components within end-to-end driving systems, ensuring interpretability, transparency, and diagnosability is vital for safety validation and failure analysis.

Current explainability techniques can be broadly categorized into two groups:

Post-hoc interpretability, which analyzes trained models using saliency methods such as Grad-CAM, guided backpropagation, or attention rollout. These techniques provide visual justifications of which modalities or spatial regions influenced a decision but often suffer from limited fidelity or inconsistent semantics.

Intrinsic interpretability, which builds explainability into the model architecture, such as through modality attribution attention (e.g., ReasonNet), causal masking, or interpretable fusion gates. These methods offer stronger semantic alignment between decisions and feature sources but may introduce architectural complexity or inference overhead.

Recent works like ThinkTwice, ReasonNet have demonstrated the feasibility of integrating explainability mechanisms into fusion pipelines, enabling users to diagnose failure cases and uncover modality reliance under diverse driving scenarios. These approaches vary in their targeted explainability dimensions—modality-level, spatial-level, or temporal reasoning, and provide complementary insights for system auditing.

Comment 5: The outlook on emerging trends such as diffusion models and LLMs remains at a conceptual level, without proposing actionable research roadmaps or evaluation metrics, making it appear rather superficial.

Response 5: We sincerely appreciate the reviewer’s suggestion regarding the depth of discussion on emerging trends. While we acknowledge that comprehensive evaluation protocols were not fully elaborated due to space limitations, we would like to clarify that the paper does incorporate non-trivial structural insights. Specifically, Figures 4 and 5 present concrete fusion frameworks involving diffusion models and Mamba-style state-space models, illustrating how these paradigms can be embedded into real-world perception and planning pipelines. Furthermore, Figure 6 offers a representative LLM-based sensor fusion design, which explores the potential of language-grounded multimodal reasoning. Rather than remaining purely conceptual, these figures are intended as architectural blueprints to inspire future research directions and serve as references for implementable system designs.

Reviewer 3 Report

Comments and Suggestions for Authors

As for multi-sensor fusion in autonomous driving, this research presents interesting findings. By the way, it needs to be revised as follows;

1) As discussed in relation to theoretical foundations and sensor characteristics, it is required to present a quantitative comparison of 2.1, 2.2, and 2.3 in tabular form, along with an analysis of the impact of their respective characteristics.

2) It is required to provide a qualitative description of the multimodal sensor data collection, preprocessing, and tuning processes at each fusion stage illustrated in Figure 1.

3) Although the architectural structure shown in Figure 3.2 is comprehensible, a clearer explanation is required regarding what the experimental results derived from this architecture are ultimately meant to demonstrate.

4) The experimental results proposed in Sections 4.1 to 4.6 of Chapter 4 should be presented.

5) It is necessary to clarify how the content presented in Chapter 5 differs from what the author previously stated. In addition, rather than merely referencing Figure 3, the specific results obtained through simulation should be clearly described. Therefore, a clear explanation of the differences is required.

6) It is recommended to provide a more detailed explanation in Chapter 6 based on the comments mentioned above.

7) In the conclusion, a detailed description is requested on what improvements have been achieved through the numerical data comparison in this study, the necessity of these improvements, and the current advantages related to multi-model fusion perception for autonomous driving.

Author Response

Responses to Reviewer 3

Comment 1: As discussed in relation to theoretical foundations and sensor characteristics, it is required to present a quantitative comparison of 2.1, 2.2, and 2.3 in tabular form, along with an analysis of the impact of their respective characteristics.

Response 1: We appreciate the reviewer’s detailed suggestion. Following the structure and intent of the comparison table in Section 2.2, we have now added two new comparison tables in Sections 2.1 and 2.3, respectively. Specifically, Table 1 summarizes the core theoretical approaches in multi-modal sensor fusion (Bayesian filtering, multi-view learning, uncertainty modeling, Transformer-based fusion), comparing their reasoning paradigm, required priors, fusion stage applicability, and limitations. Table 3 focuses on fusion methodology perspectives (complementarity modeling, probabilistic frameworks, hybrid/self-supervised learning, continual adaptation), highlighting each method’s structural assumptions, data dependence, and extendability. These additions offer a clearer cross-paradigm perspective and better support comparative analysis across foundational levels.

Table 1. Comparative Analysis of Theoretical Foundations in Sensor Fusion.

Approach

Reasoning Paradigm

Prior Knowledge

Fusion Stage

Strengths

Bayesian Filtering

Probabilistic Inference

Strong priors

Early / Middle

Handles uncertainty, recursive updates

Multi-View Learning

Representation Alignment

Weak priors

Late Fusion

Learns from view diversity

Uncertainty Modeling

Variational Approximation

Moderate priors

Middle / Late

Enhances robustness, supports risk-awareness

Transformer-based Fusion

Attention-based Structure

No explicit prior

All Stages

Captures global dependencies

Table 3. Comparative Analysis of Fusion Methodologies.

Fusion Methodology

Structural Assumptions

Data Dependency

Adaptability

Advantages

Complementarity in Fusion

Assumes informative diversity

Moderate

Task-specific

Exploits heterogeneous sensor strengths

Probabilistic Fusion Models

Likelihood-based reasoning

High

Static environments

Handles noise and ambiguity

Hybrid & Self-Supervised Approaches

Multi-loss or multi-branch

Low-to-moderate

Generalizable

Reduces labeling cost, learns semantics

Online Adaptation & Continual Learning

Temporal shift awareness

Dynamic data

High adaptability

Supports evolving environments

Comment 2: It is required to provide a qualitative description of the multimodal sensor data collection, preprocessing, and tuning processes at each fusion stage illustrated in Figure 1.

Response 2: Thank you for your valuable suggestion. We understand the importance of clearly describing the multimodal sensor data collection and preprocessing procedures associated with each fusion stage. In fact, Section 3 of our manuscript provides a comprehensive elaboration on these aspects, structured around the early, mid-level, and late fusion strategies illustrated in Figure 1.

Specifically, we highlight how the sensor data collection pipelines (e.g., camera-LiDAR synchronization, radar frame alignment, IMU integration) vary depending on the fusion stage. For early fusion, raw data are directly projected or concatenated after spatial-temporal alignment, requiring tight calibration and timestamp synchronization. In mid-level fusion, modality-specific feature extraction networks (such as ResNet for camera and sparse convolutions for LiDAR) operate independently before semantic alignment modules integrate them. In late fusion, each sensor branch undergoes full independent inference, and only final decisions are combined, with minimal preprocessing interplay.

Moreover, we detail how data preprocessing and calibration strategies are adapted to each fusion stage. For example, voxelization and depth lifting for BEV-based mid-level fusion, or feature-level attention re-weighting in Transformer-based architectures.

Thus, rather than adopting a uniform pipeline, our review emphasizes that the data handling processes are intrinsically conditioned by the chosen fusion architecture.

Comment 3: Although the architectural structure shown in Figure 3.2 is comprehensible, a clearer explanation is required regarding what the experimental results derived from this architecture are ultimately meant to demonstrate.

Response 3: We appreciate the reviewer’s thoughtful comment. As this manuscript is a review paper rather than a method proposal, the architectural illustration in Figure 2 is not intended to correspond to any single experimental result, but instead serves as a conceptual summary of representative camera–LiDAR fusion architectures widely adopted in recent literature.

Specifically, Figure 2 aims to categorize two mainstream paradigms: (1) token-based cross-modal attention mechanisms and (2) unified BEV-space fusion frameworks. These designs are decoupled from fixed outputs; rather, the downstream results depend on the selected task head attached to the shared fused representation. For instance, when integrated with a detection head, the architecture yields 3D bounding boxes; when followed by a planning head, it produces trajectory waypoints.

Comment 4: The experimental results proposed in Sections 4.1 to 4.6 of Chapter 4 should be presented.

Response 4: We thank the reviewer for this helpful suggestion. As a survey article, our intention in Chapter 4 (Sections 4.1 to 4.6) is to organize and summarize representative fusion strategies across various perception tasks such as depth completion, object detection, tracking, segmentation, and calibration. While our manuscript does not propose new experimental results of our own, we do provide quantitative performance comparisons of state-of-the-art methods as reported in the literature.

In particular, Table 6 in Section 6 presents a comparative analysis of 3D object detection results on the nuScenes test set, covering recent camera–LiDAR fusion models including TransFusion, BEVFusion, DeepInteraction, MambaFusion, and others. The table includes mean Average Precision (mAP) and nuScenes Detection Score (NDS), allowing readers to directly compare the effectiveness of different architectural choices.

Comment 5: The experimental results proposed in Sections 4.1 to 4.6 of Chapter 4 should be presented.

Response 5: We appreciate the reviewer’s detailed feedback. As a survey article, the main purpose of Chapter 5 is to provide a task-agnostic, cross-cutting evaluation of current multi-modal sensor fusion methods by synthesizing their performance limitations and structural trade-offs across multiple application domains. This differs from earlier chapters (especially Chapters 3 and 4), which primarily focused on fusion strategies and task-specific architectures (e.g., detection, segmentation, depth completion).

Specifically, Chapter 5 consolidates insights across methods and presents a broader discussion on common bottlenecks such as sensor misalignment, modality degradation, domain shift, and lack of interpretability. While earlier sections are organized by fusion stage or task, Chapter 5 addresses system-level challenges that manifest across multiple fusion pipelines, thus serving a complementary diagnostic role in the paper structure.

Regarding Figure 3, it was designed to qualitatively summarize key performance dimensions (e.g., alignment accuracy, modality robustness, generalization, interpretability, scalability, temporal consistency) based on both our own reproduction of selected benchmarks and a synthesis of reported findings in the literature.

Given the wide range of applications for multi-sensor fusion in autonomous driving, many of the challenges identified (e.g., failure under occlusion, sensor noise sensitivity) are task-dependent and difficult to isolate without dedicated ablation studies. In many cases, even visualizations may not fully reveal the root cause of performance limitations. As such, our objective in Chapter 5 is to highlight these open challenges in a principled way.

Comment 6: It is recommended to provide a more detailed explanation in Chapter 6 based on the comments mentioned above.

Response 6: We appreciate the reviewer’s suggestion to provide a more detailed explanation in Chapter 6. As a survey paper, the goal of Chapter 6 is not to reiterate prior methodological classifications, but rather to explore forward-looking research directions in multi-modal sensor fusion for autonomous driving. This chapter builds upon the system-level limitations summarized in Chapter 5, such as robustness under sensor degradation, scalability in long-horizon fusion, and interpretability in decision pipelines—and introduces three representative paradigms that are gaining increasing attention in the field. Specifically, we highlight how diffusion-based structured generation introduces stochastic denoising into BEV feature reconstruction, offering potential resilience against occlusions and missing modalities. Simultaneously, we discuss how recent state space models, such as Mamba, replace computationally intensive Transformers with lightweight recursive architectures, enabling efficient temporal modeling without sacrificing semantic consistency. Furthermore, we explore how large language models (LLMs) and vision-language frameworks are being integrated into autonomous driving stacks, enhancing task adaptability, multimodal alignment, and explainability across perception and planning. These trends are not hypothetical; rather, we support our discussion with concrete examples such as DifFUSER, MambaFusion, and LMDrive, and visualize representative pipelines in Figure 4. While the chapter does not present experimental results per se, it reflects the trajectory of cutting-edge research efforts and their potential to address existing bottlenecks. In the revised manuscript, we will further clarify this intent at the beginning of Chapter 6 and explicitly articulate its complementarity with the analyses presented in the previous sections.

Comment 7: In the conclusion, a detailed description is requested on what improvements have been achieved through the numerical data comparison in this study, the necessity of these improvements, and the current advantages related to multi-model fusion perception for autonomous driving.

Response 7: We sincerely thank the reviewer for this constructive suggestion. However, we would like to clarify that this manuscript is a survey article rather than a primary research paper presenting novel algorithms or experimental benchmarks. As such, the goal of this work is not to achieve new numerical improvements through original experiments, but rather to synthesize, compare, and contextualize existing methods in multi-modal sensor fusion for autonomous driving.

Throughout the manuscript, we provide structured analyses of representative state-of-the-art techniques, covering architectural paradigms (e.g., early/mid/late fusion, BEV representation, transformer-based attention), learning strategies (e.g., supervised, probabilistic, self-supervised), and application domains (e.g., detection, tracking, calibration). In particular, we highlight quantitative comparisons from published results, such as the benchmark analysis in Table 6 on the nuScenes dataset, to help readers understand the trade-offs between various fusion frameworks. These comparisons serve to illustrate the practical advantages (e.g., improved detection accuracy, robustness to occlusion, interpretability) achieved by recent methods under standardized evaluation settings.

Round 2

Reviewer 2 Report

Comments and Suggestions for Authors

The authors have resolved all my concerns. The manuscript is ready for publication.

Reviewer 3 Report

Comments and Suggestions for Authors

This paper has provided a satisfactory qualitative response to the previous review comments. In addition, The paper has effectively revised and elaborated on the relevant content in response it.